# Distinct pathways of homologous recombination controlled by the SWS1–SWSAP1–SPIDR complex

Rohit Prakash [1✉], Thomas Sandoval[1], Florian Morati [2], Jennifer A. Zagelbaum[3], Pei-Xin Lim [1], Travis White[1], Brett Taylor [1], Raymond Wang[1], Emilie C. B. Desclos[4], Meghan R. Sullivan[5], Hayley L. Rein[5], Kara A. Bernstein [5], Przemek M. Krawczyk[4], Jean Gautier[3], Mauro Modesti [2], Fabio Vanoli [1] & Maria Jasin [1✉]

Homology-directed repair (HDR), a critical DNA repair pathway in mammalian cells, is complex, leading to multiple outcomes with different impacts on genomic integrity. However, the factors that control these different outcomes are often not well understood. Here we show that SWS1–SWSAP1-SPIDR controls distinct types of HDR. Despite their requirement for stable assembly of RAD51 recombinase at DNA damage sites, these proteins are not essential for intra-chromosomal HDR, providing insight into why patients and mice with mutations are viable. However, SWS1–SWSAP1-SPIDR is critical for inter-homolog HDR, the first mitotic factor identified specifically for this function. Furthermore, SWS1–SWSAP1-SPIDR drives the high level of sister-chromatid exchange, promotes long-range loss of heterozygosity often involved with cancer initiation, and impels the poor growth of BLM helicase-deficient cells. The relevance of these genetic interactions is evident as SWSAP1 loss prolongs *Blm*-mutant embryo survival, suggesting a possible druggable target for the treatment of Bloom syndrome.

[1] Developmental Biology Program, Memorial Sloan Kettering Cancer Center, New York, NY, USA. [2] Cancer Research Center of Marseille, CNRS, Inserm, Institut Paoli-Calmettes, Aix-Marseille Université, Marseille, France. [3] Department of Genetics and Development and Institute for Cancer Genetics, College of Physicians and Surgeons, Columbia University, New York, NY, USA. [4] Department of Medical Biology, Amsterdam University Medical Centers, Amsterdam, The Netherlands. [5] Department of Microbiology and Molecular Genetics, UPMC Hillman Cancer Center, University of Pittsburgh School of Medicine, Pittsburgh, PA, USA. ✉email: rohitpraka@gmail.com; m-jasin@ski.mskcc.org

D ouble-strand breaks (DSBs) are among the most danger-ous DNA lesions that can arise in cells and, if not repaired correctly, can lead to genomic instability and tumorigenesis[1]. Homologous recombination, also known as homology-directed repair (HDR), is the predominant pathway to repair DSBs in an error-free manner. The RAD51 recombinase has a central role, whereby it forms filaments on single-stranded DNA at resected DNA ends through the activity of mediator proteins like BRCA2[2] and can be stabilized and/or remodeled by RAD51 paralogs, as shown for the yeast and worm proteins[3,4]. RAD51 nucleoprotein filaments subsequently invade a homo-logous template, typically the sister chromatid, to form a dis-placement loop (D loop), followed by repair synthesis. DNA helicases can unwind these D loops to promote HDR by the synthesis-dependent strand annealing pathway to result in non-crossovers. Alternatively, D loops that capture the other DNA end can mature into double Holliday junctions, which can be dis-solved by BLM, a RECQ helicase deficient in individuals with Bloom syndrome, or resolved by structure-specific nucleases to form crossovers[5,6]. What controls the decision points for these various pathways is not well understood.

SWS1–SWSAP1 is a recently identified complex that is criti-cally important for promoting the stable assembly of both RAD51 and DMC1 nucleoprotein filaments at resected DNA ends during meiosis[7]. SWS1 homologs are readily identified in a number of organisms based on the conservation of the zinc-coordinating domain ($CxC(x_{15})CxH$ in mice and humans; Supplementary Fig. 1a)[8,9]. SWSAP1 has limited homology to RAD51, includ-ing at Walker A and B motifs predicted to be important for ATPase activity, and so it is considered to be a RAD51 paralog (Supplementary Fig. 1b)[10]. SWSAP1 has the RAD51-interacting motif FxxA and has been shown to bind RAD51[11]. Moreover, SWSAP1 also interacts with SPIDR, a potential scaffolding pro-tein with no identified protein motifs (Supplementary Fig. 1c), but which has also been shown to interact with RAD51[12,13]. SPIDR was initially identified in human cells as a binding partner of BLM and was reported to act in the same pathway as BLM in Hela cells[12]. SWS1-containing complexes, often termed "Shu" complexes, vary substantially in different organisms with regards to the number and structure of the other protein components, although like SWSAP1, they typically show limited homology to RAD51 (Supplementary Fig. 1d)[14,15].

Budding and fission yeast Shu complex mutants display a number of HDR-related phenotypes[14], including meiotic defects, which are relatively mild compared with mouse mutants[16–18]. Notably, yeast Shu mutants can suppress the severe phenotypes associated with deficiency of the DNA helicases Sgs1/Rqh1 (BLM homologs in budding/fission yeasts) and Srs2, for example, the severe DNA damage sensitivity of sgs1/rqh1 mutants in both yeasts and the srs2 mutant in fission yeast[8,19,20]. Biochemical studies have demonstrated that the four-member budding yeast Shu complex works together with the canonical Rad51 paralog complex Rad55-Rad57 and Rad52 to promote assembly of Rad51 on single-stranded DNA (ssDNA) pre-coated with replication protein A (RPA)[21].

A role for the human SWS1–SWSAP1–SPIDR complex in mitotic HDR has been suggested in studies in human cell lines depleted for individual complex members, which show reduced RAD51 focus formation and mild sensitivity to DNA damaging agents[8,10,13]. Biochemical experiments with SWSAP1 have indi-cated that it stabilizes RAD51 filaments by blocking the activity of the helicase FIGNL1[11], supporting the in vivo observations. However, the types of HDR promoted by SWS1–SWSAP1–SPIDR and the genetic interaction with BLM have not been delineated. Moreover, the diversity of the Shu complex composition in dif-ferent organisms precludes a direct comparison.

In this study, we show that mouse SWS1, SWSAP1, and SPIDR impact multiple types of HDR. Although not required for HDR involving repeats on the same chromosome or gene targeting, these proteins play a crucial role in inter-homolog homologous recombination (IH-HR). This specific defect in IH-HR contrasts with the more general HDR defect observed upon interfering directly with RAD51 function, in which all types of HDR are reduced. Sister-chromatid exchanges (SCEs) induced by methyl methanesulfonate (MMS) or inhibition of poly(ADP-ribose) polymerases are partially reduced in Sws1, Swsap1, and Spidr mutants. More strikingly, SCEs generated by BLM loss are nearly totally abrogated in these mutants, whereas other HDR events are actually increased with complex disruption. Concomitantly, the loss of SWSAP1 in a Blm mouse mutant prolongs its embryonic survival. We propose that SWS1–SWSAP1–SPIDR activity has an important role early in the HDR pathway in determining the fate of the recombination intermediates, where it functions specifically to stabilize intermediates, which are processed by BLM.

## Results

**SWS1, SWSAP1, and SPIDR form a complex**. Human SWS1, SWSAP1, and SPIDR interact with each other in yeast three-hybrid assays[8,10–13]. We tested these interactions with purified proteins, co-expressing SWS1 and SWSAP1 since SWS1 was not stable on its own in bacteria. SWS1–SWSAP1 was found to bind to SPIDR, demonstrating that these three proteins interact with each other as the SWS1–SWSAP1–SPIDR complex (Fig. 1a). Given that SPIDR was initially identified as a BLM interacting protein[12], we asked whether SWS1–SWSAP1 could also interact with BLM and observed an interaction using purified proteins (Supplementary Fig. 2a). We tested whether interactions reported for human proteins also occur for the mouse proteins, finding that mouse SWSAP1 interacts with SWS1 and SPIDR (Supple-mentary Fig. 2b–d). In addition, we observed that mouse SWS1–SWSAP1 and SPIDR, like the human proteins[10–12,20], interact with RAD51, as well as with DMC1, the meiosis-specific RAD51 homolog (Supplementary Fig. 2e–j).

**Like SWS1–SWSAP1, SPIDR is not essential in the mouse but is required during meiosis**. $Sws1^{-/-}$ and $Swsap1^{-/-}$ mice are viable[7], unlike mutants in many other HDR genes[22]. The inter-action with SWS1–SWSAP1 led us to ask whether loss of SPIDR in mice would phenocopy loss of SWS1–SWSAP1 or whether it would be essential like BLM[23]. To test this, we disrupted the gene in fertilized mouse eggs (Supplementary Fig. 3a). As with $Sws1^{-/-}$ and $Swsap1^{-/-}$ but unlike $Blm^{-/-}$ mice, $Spidr^{-/-}$ mice are born at the expected Mendelian ratio and have a normal adult body weight (Supplementary Fig. 4a–c), demonstrating that SPIDR is not essential for development.

The most pronounced phenotype of $Sws1^{-/-}$ and $Swsap1^{-/-}$ mice is the reduced size of their gonads[7]. Similar to these mutants, testes, and ovaries of $Spidr^{-/-}$ mice are about one-third the size of control mice (Supplementary Fig. 4b, c). Seminiferous tubules have greatly reduced cellularity owing to a paucity of post-meiotic germ cells, indicating defective meiosis (Fig. 1b and Supplemen-tary Fig. 4d). Furthermore, as with $Sws1^{-/-}$ and $Swsap1^{-/-}$, meiotic RAD51 and DMC1 focus formation are reduced ~three-fold in $Spidr^{-/-}$ spermatocytes (Fig. 1c, d; Supplementary Fig. 4e–h). The presence of a few post-meiotic cells, including elongating spermatids (Fig. 1b), indicates a slightly milder meiotic phenotype than $Sws1^{-/-}$ and $Swsap1^{-/-}$, although it remains possible that the mutant Spidr allele retains some function. Thus, although SPIDR is not required for mouse embryonic develop-ment, it is critical for RAD51 and DMC1 focus formation during spermatogenesis, similar to SWS1 and SWSAP1[7].

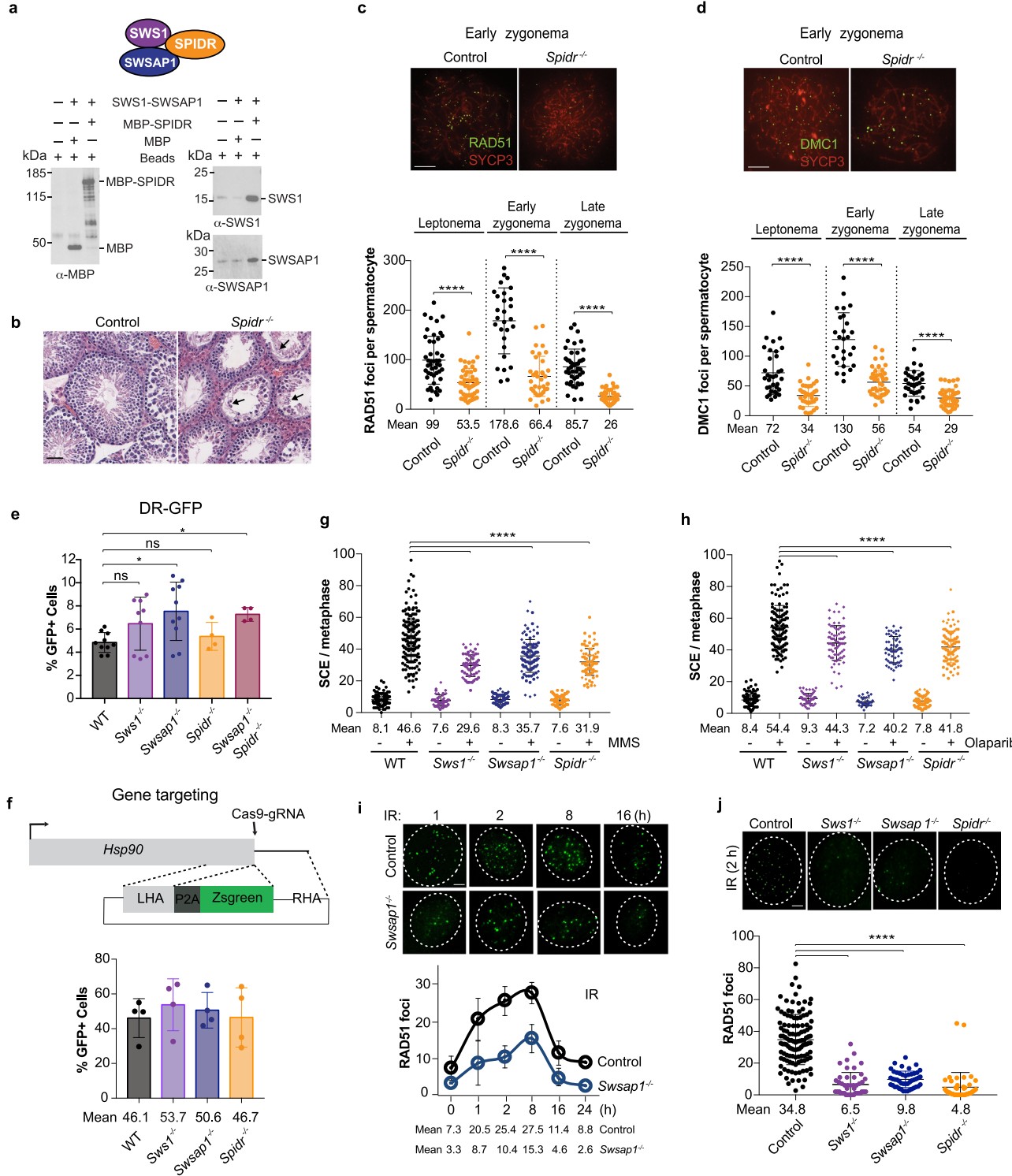

**HDR in mitotic cells does not require observable RAD51 focus formation**. The viability and normal body weights seen for $Sws1^{-/-}$, $Swsap1^{-/-}$, and $Spidr^{-/-}$ mice led us to question whether SWS1–SWSAP1–SPIDR is required for HDR in mitotic cells, given the embryonic lethality observed for other HDR mutants. To directly measure HDR, we utilized the commonly employed reporter DR-GFP, which measures HDR without crossing over between direct repeats (intrachromatid or sister chromatid) leading to GFP-positive cells following I-SceI endonuclease expression[24]. We first measured HDR in mouse embryonic stem (ES) cells, which have a high S phase

component and thus may rely heavily on HDR. Using CRISPR-Cas9, we disrupted $Sws1$, $Swsap1$, and $Spidr$ in ES cells containing an integrated DR-GFP reporter (Supplementary Figs. 3b and 5a). Testing multiple mutant ES cell lines for each gene, we found that single and double mutant cells are proficient in HDR (Fig. 1e and Supplementary Fig. 6a). The increase in HDR in $Swsap1^{-/-}$ cells is likely due to clonal variation, as expressing SWSAP1 did not affect HDR levels (Supplementary Fig. 6a, right).

We also quantified HDR in primary ear fibroblasts from $Sws1^{-/-}$ and $Swsap1^{-/-}$ mice containing the DR-GFP reporter[25]. As with ES

**Fig. 1 SWS1–SWSAP1–SPIDR promotes distinct types of HDR. a** SWS1, SWSAP1, and SPIDR form a complex. SPIDR was pulled down with anti-MBP beads and interacting proteins were identified by western blotting with the indicated antibodies. A schematic of the complex is shown on top, the color code for the individual components is maintained in subsequent figures. **b** Testis section from a *Spidr* mutant showing seminiferous tubules with substantially reduced post-meiotic germ cells. Arrows indicate infrequent round and elongating spermatids. Scale bar, 50 μm. **c, d** Representative chromosome spreads from adult mice from early zygonema from control and *Spidr* mutant spermatocytes to analyze RAD51 and DMC1 focus formation. Foci were counted on the developing chromosome axes that are marked by SYCP3, which at early zygonema form short linear stretches. Scale bar, 10 μm. RAD51 (**c**) and DMC1 (**d**) focus formation are substantially reduced in *Spidr*$^{-/-}$ spermatocytes at early meiotic prophase stages. $n = 2$. (See also Supplementary Fig. 4e–h). **e** *Sws1*$^{-/-}$, *Swsap1*$^{-/-}$, *Spidr*$^{-/-}$, and *Swsap1*$^{-/-}$*Spidr*$^{-/-}$ ES cells are proficient in HDR as measured with the DR-GFP reporter. WT $n = 14$, *Sws1*$^{-/-}$ $n = 9$, *Swsap1*$^{-/-}$ $n = 11$, *Spidr*$^{-/-}$ $n = 4$, and *Swsap1*$^{-/-}$*Spidr*$^{-/-}$ $n = 4$, where $n$ is the number of independent experiments. (See also Supplementary Fig. 6a). **f** *Sws1*$^{-/-}$, *Swsap1*$^{-/-}$, and *Spidr*$^{-/-}$ ES cells are proficient at DSB-induced gene targeting. Top, schematic of the assay[26]. A DSB is introduced at the terminus of the *Hsp90* gene by Cas9-gRNA. Upon integration of the ZsGreen-coding region through the left and right homology arms (LHA and RHA, respectively) in the circular plasmid, ZsGreen is expressed. Bottom, Gene targeting levels. Values are 4 days after expression of Cas9-gRNA; in the absence of Cas9-gRNA, GFP+ cells are <0.2%. $n = 3$, where $n$ is the number of independent experiments. **g, h** SCEs are reduced with *Sws1*, *Swsap1*, and *Spidr* mutation in ES cells after treatment with MMS (0.5 mM for 1 h) (**g**) and olaparib (20 nM for 17 h) (**h**). Two independent clones are tested for each mutant, as differentiated by diamonds and circles. **g** WT-MMS $n = 5$, WT +MMS $n = 6$, *Sws1*$^{-/-}$ -MMS $n = 4$, *Sws1*$^{-/-}$ +MMS $n = 5$, *Swsap1*$^{-/-}$ -MMS $n = 4$, *Swsap1*$^{-/-}$ +MMS $n = 5$, *Spidr*$^{-/-}$ -MMS $n = 3$, *Spidr*$^{-/-}$ +MMS $n = 3$. **h** WT-olaparib $n = 6$, WT + olaparib $n = 7$, *Sws1*$^{-/-}$ -olaparib $n = 3$, *Sws1*$^{-/-}$ +olaparib $n = 4$, *Swsap1*$^{-/-}$ -olaparib $n = 3$, *Swsap1*$^{-/-}$ +olaparib $n = 4$, *Spidr*$^{-/-}$ -olaparib $n = 2$, *Spidr*$^{-/-}$ +olaparib $n = 2$, where $n$ is the number of independent experiments. (See also Supplementary Fig. 6c, d). **i** RAD51 focus formation in *Swsap1*$^{-/-}$ primary ear fibroblasts is reduced two- to three-fold, but has similar kinetics as control cells when treated with 10 Gy IR. Scale bar, 10 μm. $n = 3$, where $n$ is the number of independent experiments. **j** *Sws1*$^{-/-}$, *Swsap1*$^{-/-}$, and *Spidr*$^{-/-}$ primary ear fibroblasts have reduced RAD51 focus formation compared with the control cells upon exposure to IR (10 Gy, 2 h). Scale bar, 10 μm. $n = 3$, where $n$ is the number of independent experiments. Error bars in **e–j**, mean ± s.d. *$P \leq 0.05$; **$P \leq 0.01$; ***$P \leq 0.001$; ****$P \leq 0.0001$; unpaired $t$ test, two-tailed. All source data are provided in the source data file.

cells, we found that HDR levels are similar to controls, as measured directly by the fraction of GFP positive cells or taking into account total I-SceI site-loss (Supplementary Fig. 6b).

HDR was also examined in mutant ES cells using a gene targeting assay. A DSB was introduced at the *Hsp90* locus by CRISPR-Cas9 to induce repair from a transfected promoterless ZsGreen template flanked by *Hsp90* homology arms[26] (Fig. 1f). As in the direct-repeat assay, gene targeting levels are similar in wild-type and *Sws1*$^{-/-}$, *Swsap1*$^{-/-}$, and *Spidr*$^{-/-}$ mutant cells (Fig. 1f). Thus, SWS1–SWSAP1–SPIDR is not required for either intrachromosomal HDR or HDR with an extrachromosomal template.

SCEs provide a distinct measure of homologous recombination. In wild-type cells, SCEs are increased substantially by exposure to either MMS or the poly(ADP-ribose) polymerase inhibitor olaparib (approximately six-fold for each; Fig. 1g, h). SCEs are also induced in *Sws1*$^{-/-}$, *Swsap1*$^{-/-}$, and *Spidr*$^{-/-}$ cells, but at levels ~30% lower on average than in wild-type (Fig. 1g, h and Supplementary Fig. 6c, d). Thus, these agents induce SCEs in mutant cells but the induction is mildly reduced compared to wild-type cells.

The lack of an observable HDR defect with SWS1–SWSAP1–SPIDR deficiency in some assays (DR-GFP, gene targeting) and minimal effect in another (SCE) is in line with the viability of mutant mice, but is surprising given reports from knockdown experiments in human cell lines that depletion of these proteins reduces RAD51 focus formation following exposure to DNA-damaging agents[10]. Therefore, we monitored the kinetics of RAD51 focus formation in *Swsap1*$^{-/-}$ primary ear fibroblasts after ionizing radiation (IR). Although DNA damage was found to be similar, as gauged by γH2AX (Supplementary Fig. 6f), we observe reduced RAD51 foci at each time point tested (Fig. 1i). We then examined fibroblasts from *Sws1*$^{-/-}$, *Swsap1*$^{-/-}$, and *Spidr*$^{-/-}$ mice at 2 hr post IR, when RAD51 foci are high in wild-type. In each mutant, a substantial reduction in RAD51 foci is observed following exposure to IR (four- to nine-fold) (Fig. 1j). RAD51 focus formation is also greatly reduced following exposure to MMS (8–10-fold) in *Sws1*$^{-/-}$ and *Swsap1*$^{-/-}$ cells (Supplementary Fig. 6e). By contrast, RPA focus formation is similar (Supplementary Fig. 6g), indicating that DNA end resection, which is a prerequisite for RAD51 loading, is not aberrant. Thus,

observable RAD51 foci are not required for cellular HDR proficiency.

**SWS1–SWSAP1–SPIDR promotes efficient IH-HR.** Thus far, mitotic components specifically required for HDR between homologous chromosomes, termed IH-HR, have not been identified. Given that SWS1, SWSAP1, and SPIDR are required during meiosis, we considered whether these proteins play a role in IH-HR in mitotic cells, even if their contribution to intrachromosomal HDR appears to be minimal. IH-HR is critical during meiosis, occurring while chromosomes undergo massive structural changes during synapsis while requiring meiotic-specific proteins like DMC1[27]. By contrast, in mitotic cells, the sister chromatid is the preferred template for HDR. Although the homolog can serve as a template for repair in mitotic cells, it is used at low frequency[28] and in the absence of chromosome synapsis.

To determine whether SWS1–SWSAP1–SPIDR has a role in IH-HR, we used 129/B6 ES cells carrying the S/P reporter[29,30]. In brief, two nonfunctional neomycin (*neo*) genes are present at allelic positions on chromosome 14: one is nonfunctional due to the insertion of an I-SceI endonuclease site (S allele) and the other due to the insertion of a PacI restriction site (P allele) (Fig. 2a). Once a DSB is generated by I-SceI endonuclease in the S allele, repair by IH-HR using the P allele as a template results in a *neo*$^+$ recombinant. Most IH-HR events are simple gene conversions not involving a crossover, although a fraction is crossovers which are increased by the loss of BLM[30].

We examined S/P reporter cells mutated for each member of the SWS1–SWSAP1–SPIDR complex (Supplementary Figs. 3c and 5b). The recovery of *neo*$^+$ recombinants is reduced five to six-fold in each of the *Sws1*$^{-/-}$, *Swsap1*$^{-/-}$, and *Spidr*$^{-/-}$ cell lines relative to wild-type (Fig. 2b and Supplementary Fig. 7a), indicating defective IH-HR. Complementation experiments confirmed that the IH-HR defects are owing to loss of the cognate proteins. Similar to single mutant cells, *Swsap1*$^{-/-}$ *Spidr*$^{-/-}$ cells show a five-fold reduction in IH-HR, which can be restored only by coexpression of both SWSAP1 and SPIDR (Fig. 2b).

The requirement for SWS1–SWSAP1–SPIDR for IH-HR was unanticipated, given that these proteins are not required for intrachromosomal HDR. After DNA replication, sister

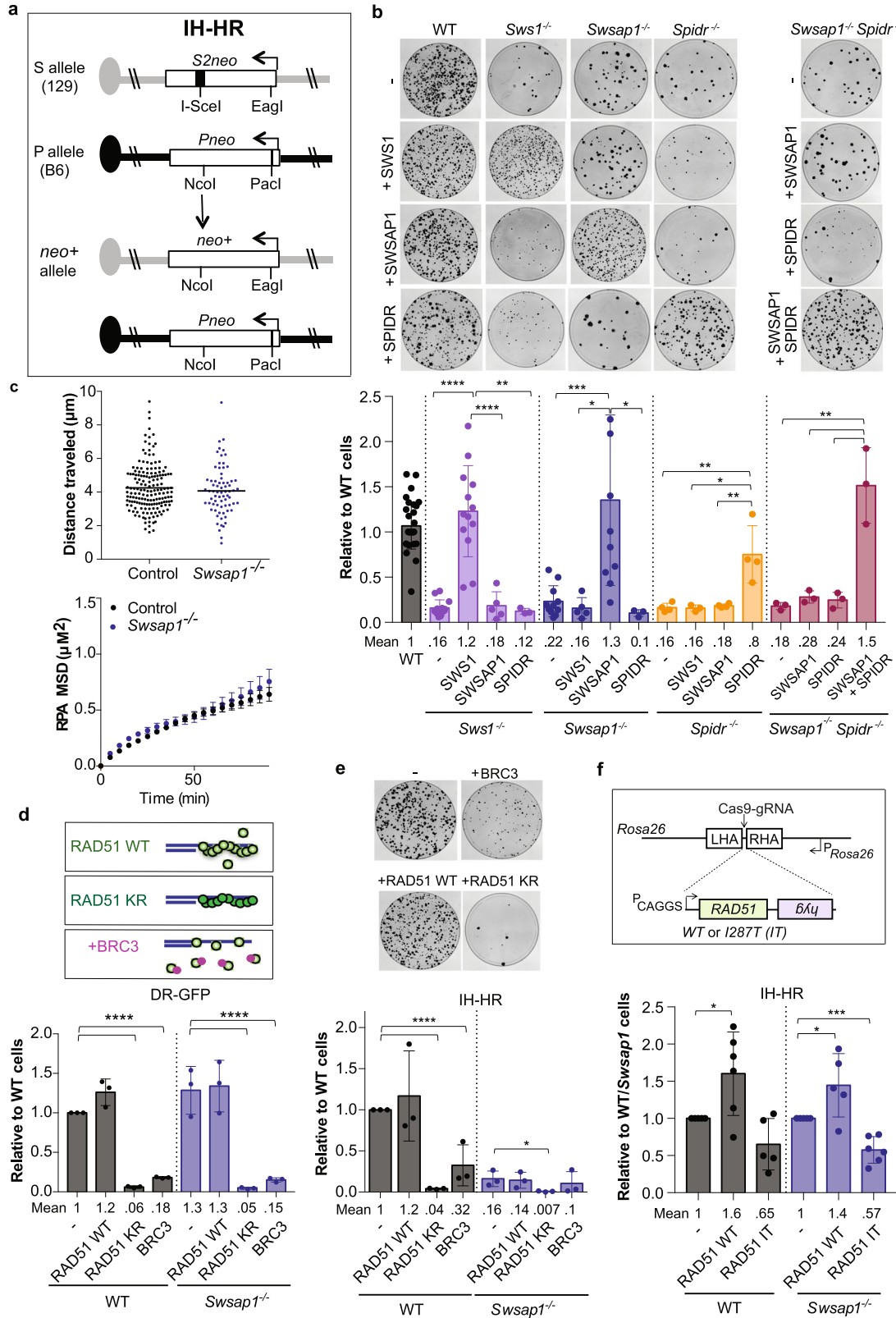

chromatids are held in proximity as long as cohesion is maintained; however, homologs are not paired, such that IH-HR may require a more extensive homology search process. We asked, therefore, whether the mobility of DNA ends is affected in cells deficient for SWS1–SWSAP1–SPIDR after DSB formation. Immortalized mouse embryonic fibroblasts (MEFs) were treated with neocarzinostatin (NCS) to induce DSBs, and the movement

of GFP-tagged single-stranded binding protein RPA was monitored over time[31]. We found that the cumulative distance traveled by RPA-GFP was similar in control and *Swsap1*$^{-/-}$ MEFs (Top panel, Fig. 2c). The mean-square displacement is also similar (bottom panel, Fig. 2c), indicating that DNA end mobility is not significantly altered by loss of SWSAP1, and so is unlikely to account for reduced IH-HR.

**Fig. 2 The SWS1–SWSAP1–SPIDR complex promotes IH-HR. a** IH-HR at the S/P reporter in 129/B6 hybrid ES cells. The S allele on the 129 chromosome 14 contains the *S2neo* gene mutated by insertion of an I-SceI cleavage site at a NcoI restriction site, whereas the P allele on the B6 chromosome 14 contains the *Pneo* gene mutated by insertion of a PacI restriction site at an EagI site. After induction of an I-SceI-induced DSB in *S2neo*, IH-HR using the Pneo gene as a template results in a functional neomycin gene (*neo*+) that gives rise to G418 resistant colonies. **b** *Sws1*−/−, *Swsap1*−/−, *Spidr*−/−, and *Swsap1*−/− *Spidr*−/− cells show decreased numbers of *neo*+ colonies, indicative of reduced IH-HR, which is rescued by expression of the cognate proteins. Representative Giemsa-stained plates in S/P reporter cells after I-SceI expression are shown at the top with quantification below. Relative to WT cells, colony counts expressed relative to wild-type cells transfected with an empty vector within each experiment. WT $n = 36$, *Sws1*−/− $n = 14$, *Sws1*−/− +SWS1 $n = 14$, *Sws1*−/− +SWSAP1 $n = 6$, *Sws1*−/− +SPIDR $n = 3$, *Swsap1*−/− $n = 12$, *Swsap1*−/− +SWS1 $n = 6$, *Swsap1*−/− +SWSAP1 $n = 12$, *Swsap1*−/− +SPIDR $n = 3$, *Spidr*−/− $n = 5$, *Spidr*−/− +SWS1 $n = 3$, *Spidr*−/− +SWSAP1 $n = 5$, *Spidr*−/− +SPIDR $n = 5$, *Swsap1*−/−*Spidr*−/− $n = 3$, *Swsap1*−/−*Spidr*−/− +SWSAP1 $n = 3$, *Swsap1*−/−*Spidr*−/− +SPIDR $n = 3$, and *Swsap1*−/−*Spidr*−/− +SWSAP1 + SPIDR $n = 3$, where $n$ is the number of independent experiments. (See also Supplementary Fig. 7a.). **c** DNA end mobility is not significantly altered by loss of SWSAP1. MEFs were treated with NCS (0.5 μg/mL) for 100 min and RPA-GFP foci (RPA32-GFP) were tracked. Top, the mean cumulative distance traveled by RPA-GFP foci in control and *Swsap1*-defective MEFs is similar. Bottom, The mean-square displacement (MSD) of RPA-GFP foci following NCS treatment is also similar: Control, $4.217 \times 10^{-5}$ μm² s⁻¹; *Swsap1*−/−, $5.051 \times 10^{-5}$ μm² s⁻¹. $n = >450$ foci in >8 cells. Error bars, weighted S.E.M. overall MSD curves. **d, e** RAD51 K133R (KR) and BRC3 peptide expression reduce HDR in both the DR-GFP (**d**) and IH-HR (**e**) reporters in wild-type cells. In *Swsap1*−/− cells, RAD51 K133R (KR) and BRC3 peptide expression reduces HDR in DR-GFP to a similar extent as in wild-type cells (**d**), whereas it further reduces IH-HR (**e**). RAD51 KR is overexpressed relative to endogenous RAD51 and forms filaments with slow turnover because of the ATP hydrolysis defect. BRC3 peptides can bind RAD51 and sequester it from forming nucleoprotein filaments. Relative to WT cells, % GFP+ or colony counts expressed relative to wild-type cells transfected with an empty vector within each experiment. For **d, e**, $n = 3$, where $n$ is the number of independent experiments. **f** Constitutive overexpression of RAD51 WT increases IH-HR in wild-type and *Swsap1*−/− cells, whereas RAD51 I287T expression reduces IH-HR. WT $n = 5$, WT + RAD51 WT $n = 6$, WT + RAD51IT $n = 5$, *Swsap1*−/− $n = 5$, *Swsap1*−/− +RAD51 WT $n = 5$, *Swsap1*−/− +RAD51IT $n = 6$, where $n$ is the number of independent experiments. An expression cassette for RAD51 WT or RAD51 I287T was targeted to *Rosa26* locus[81] in wild-type and *Swsap1*−/− cells, selecting for expression of the *hyg* gene from the *Rosa26* promoter. Two targeted clones for each were analyzed in IH-HR assays. Relative to WT/*Swsap1*−/− cells, colony counts expressed relative to wild-type or *Swsap1*−/− cells (i.e., WT/*Swsap1* cells on the y-axis) transfected with an empty vector (–) within each experiment. (See also Supplementary Fig. 7e, f) Error bars in **b**, **d–f** mean ± s.d. *$P \leq 0.05$; **$P \leq 0.01$; ***$P \leq 0.001$; ****$P \leq 0.0001$; unpaired $t$ test, two-tailed. All source data are provided in the source data file.

**Specific role of SWS1–SWSAP1–SPIDR in IH-HR is distinct from core HDR factors**. The substantial reduction in IH-HR in *Sws1*−/−, *Swsap1*−/−, and *Spidr*−/− mutant cells contrasts with the lack of an observable defect in the DR-GFP reporter assay, which involves repair from the sister or same chromatid. We tested whether other factors would show differential effects on HDR in these two contexts. To this end, we transiently expressed a mutant form of RAD51 or a peptide from BRCA2, both of which have previously been shown to reduce HDR in the DR-GFP reporter through different mechanisms, i.e., RAD51 K133R, which forms hyper stable RAD51 nucleoprotein filaments owing to a defect in ATP hydrolysis[32,33], and the BRC3 repeat from BRCA2, which sequesters RAD51 to impede RAD51 filament formation (Fig. 2d)[32,34,35]. RAD51 K133R expression substantially reduces both HDR in the DR-GFP reporter (Fig. 2d) and IH-HR (Fig. 2e). BRC3 expression also reduces HDR in both assays, although to a lesser extent. Thus, RAD51 K133R or BRC3 expression affects both types of HDR in contrast to the specific reduction of IH-HR in *Sws1*−/−, *Swsap1*−/−, and *Spidr*−/− cells.

We also tested RAD51 K133R and BRC3 expression in *Sws1*−/− and *Swsap1*−/− cells and found that HDR in the DR-GFP assay is impacted similarly to that of wild-type cells, indicating that SWS1 and SWSAP1 do not function in this type of HDR, even as a "backup" (Fig. 2d and Supplementary Fig. 7b). However, both RAD51 K133R and BRC3 further reduce IH-HR in *Swsap1*−/− cells (Fig. 2e), consistent with a distinct role for the complex in specific types of HDR.

Given that SWS1–SWSAP1–SPIDR mutant cells show defects in RAD51 focus formation, we considered whether impaired RAD51 nucleoprotein filament assembly/stability is an underlying cause of the IH-HR defects in mutant cells. The severe reduction in HDR with RAD51 K133R expression, which leads to hyperstable filaments, however, points to the importance of turnover of the RAD51 nucleoprotein filament to complete HDR reactions. Another variant of RAD51 has been identified in budding yeast (I345T), which modulates the stability of RAD51 nucleoprotein filaments through another mechanism by enhancing ssDNA binding[36,37]. Notably, this mutant partially suppresses the recombination defects of budding yeast RAD51 paralog mutants *rad55* and *rad57*[36]. We tested the cognate mammalian mutant, RAD51 I287T, to determine whether it could rescue the IH-HR defects of *Swsap1*−/− cells. However, transient expression of RAD51 I287T reduces IH-HR in both wild-type and mutant cells, as well as HDR in DR-GFP, although not to the same extent as RAD51 K133R (Supplementary Fig. 7c, d) suggesting again that slower turnover of RAD51 filaments is inhibitory to HDR.

Since transient transfection leads to high levels of protein expression, we also tested constitutive expression from the *Rosa26* locus, which leads to similar protein levels as from the endogenous locus, approximately doubling the total RAD51 in the cell (Supplementary Fig. 7e). In this case too, RAD51 I287T expression reduces IH-HR in both wild-type and *Swsap1*−/− cells (Fig. 2f and Supplementary Fig. 7f). In contrast, however, constitutive RAD51 WT overexpression results in a moderate increase in IH-HR in both wild-type (1.6-fold) and *Swsap1*−/− (1.4-fold) cells. These results suggest that elevated RAD51 WT levels lead to more stable or longer RAD51 filaments that promote IH-HR. Because an increase in IH-HR is not observed when RAD51 WT is highly expressed (Fig. 2e), the level of RAD51 is likely to be finely tuned for efficient HDR.

**SWSAP1 requirements to promote IH-HR**. Sequence alignment of mouse and human SWSAP1 reveals two notable differences (Supplementary Fig. 1b): the mouse protein contains a 21-amino acid N-terminal extension, which although not annotated, can also be found encoded within human exon 1 after alternative translation initiation. In addition, although the human protein has been reported to support ATPase activity involving canonical Walker A and B motifs[10], the mouse protein lacks critical residues within the Walker A motif (human: GKT vs. mouse: AQT). Given these distinctions, we examined the ability of human SWSAP1 to cross complement the mouse ES cell mutants using the IH-HR assay. Although human SWS1 complements *Sws1*−/− cells for IH-HR as efficiently as mouse SWS1 (Supplementary Fig. 8a), the annotated human SWSAP1—the "short" form—only partially

complements $Swsap1^{-/-}$ cells (Supplementary Fig. 8b). However, the "long" form of human SWSAP1 containing the N-terminal extension complements as efficiently as mouse SWSAP1. Thus, the N terminus of SWSAP1 is necessary for full activity, although it does not affect the interaction with either SWS1 or RAD51 (Supplementary Fig. 8c, d).

The canonical RAD51 paralogs have variable requirements for residues involved in ATP binding/hydrolysis, in some cases requiring both the Walker A and B motifs and in other cases only the Walker B motif[22]. To examine the Walker motifs of mouse SWSAP1, we mutated a conserved residue within the Walker B motif (D115A), as well as a non-canonical residue within the Walker A motif (Q37A) and tested them in the IH-HR assay (Supplementary Fig. 8e). Both SWSAP1 Q37A and D115A are able to substantially complement the IH-HR defects of $Swsap1^{-/-}$ cells, such that only modest defects are observed. We also tested the contributions of the human SWSAP1 Walker A and B motifs and found that Walker A (K39A, K39R) and Walker B (D117A) mutants are as active as the wild-type protein (Supplementary Fig. 8f). An exchange of the canonical Walker A motif in the human protein with the non-canonical mouse motif ("Swap") is also functional. As a comparison, we also tested human and mouse SWSAP1 Walker A and B mutants in wild-type and $Swsap1^{-/-}$ cells containing the DR-GFP reporter, finding that human and mouse SWSAP1 Walker A and B mutants are proficient also in this type of HDR (Supplementary Fig. 8g). Thus, canonical Walker motifs are apparently not required for SWSAP1 function in HDR.

A RAD51 interaction motif, FxxA, has been identified in several proteins[38,39], including recently in human SWSAP1 (Supplementary Fig. 1b), such that mutation of this motif from FAAA to EAAE impairs interaction with RAD51[11]. The aromatic ring of the phenylalanine in this motif in a BRCA2 peptide is buried within a hydrophobic cavity of RAD51 in the crystal structure[38], suggesting that it may be particularly important. We found that mutation to EAAA in either mouse or human SWSAP1 abrogates its ability to rescue the IH-HR defects of $Swsap1^{-/-}$ cells and to the same extent as mutation to EAAE (Supplementary Fig. 8e, f), highlighting a key role for the phenylalanine within this motif.

**SWS1–SWSAP1–SPIDR drives sister chromatid exchange and poor growth in the absence of BLM.** Cells deficient in the BLM helicase have greatly elevated levels of SCEs[40] which are dependent on HDR factors[41]. Given that DNA damage-induced SCEs are marginally reduced in the absence of SWS1–SWSAP1–SPIDR, we asked whether SCE induction by genetic perturbation of BLM has similar consequences. The 129/B6 ES cells contain modified $Blm$ alleles ($Blm^{tet/tet}$), such that doxycycline (Dox) addition results in depletion of BLM[42]. Upon Dox addition, these cells experience an ~nine-fold induction of SCEs (Fig. 3a and Supplementary Fig. 9a), such that the mean number of SCEs is ~100 per cell, comparable to what is observed in Bloom syndrome patient cells[43]. Remarkably, BLM depletion in $Sws1^{-/-}$ $Blm^{tet/tet}$, $Swsap1^{-/-}$ $Blm^{tet/tet}$, or $Spidr^{-/-}$ $Blm^{tet/tet}$ ES cells (Supplementary Figs. 3c and 5b) reduces levels of SCEs to nearly normal levels (Fig. 3a and Supplementary Fig. 9a), indicating that SWS1–SWSAP1–SPIDR drives SCE formation in the absence of BLM. The profound reduction in SCEs upon depletion of BLM over the moderate reduction in SCEs by either MMS or olaparib in mutant cells indicates a preference for SWS1–SWSAP1–SPIDR to act in pathways involving distinct types of lesions.

IH-HR in mitotic cells typically leads to only local changes at the site of repair. However, when crossing over occurs during IH-HR, it runs the risk of generating loss of heterozygosity (LOH)

from the site of the crossover to the telomere (i.e., long-range LOH), which can contribute to cancer initiation[44,45]. In addition to SCEs, crossing over between homologs is also suppressed by BLM[30]. We found that depletion of BLM does not further decrease overall IH-HR frequencies in $Sws1^{-/-}$, $Swsap1^{-/-}$, or $Spidr^{-/-}$ cells (Fig. 3b and Supplementary Fig. 9b). Interestingly, crossing over either with or without BLM depletion is reduced by $Sws1$ or $Swsap1$ mutation, as indicated by fewer colonies with LOH of a distal marker. Thus, deleterious LOH that arises upon IH-HR is substantially reduced by loss of SWS1–SWSAP1–SPIDR through the combined reduction in IH-HR frequency and fewer LOH outcomes.

To further examine the relationship between SWS1–SWSAP1–SPIDR and BLM, we used DR-GFP reporter and $Hsp90$ gene targeting assays in ES cells treated with the BLM inhibitor ML216 or with Dox to deplete BLM, respectively. With BLM deficiency, we observed a small reduction in HDR in the DR-GFP assay, consistent with previous results[46], and also in the gene targeting assay (Fig. 3c, d). Notably, concomitant loss of SWSAP1 leads to a partial restoration of HDR in both assays. We also tested RAD51 focus formation in $Swsap1$ mutant MEFs treated with a $Blm$ shRNA (Supplementary Fig. 9c) and found a partial restoration of RAD51 foci upon loss of BLM (Fig. 3e).

BLM has been reported to affect cell proliferation. Primary human and mouse fibroblasts deficient in BLM grow slowly[23,47,48] and depletion of BLM in $Blm^{tet/tet}$ ES cells reduces colony formation[30]. Remarkably, loss of SWS1, SWSAP1, or SPIDR significantly rescues colony formation upon BLM depletion, such that the number of colonies and their size are substantially restored (Fig. 3f and Supplementary Fig. 9d). Further, population doubling is also restored. At passages 2 and 3, BLM-depleted ES cells show a substantial reduction in cell number compared to non-depleted cells, whereas depletion of BLM in $Sws1^{-/-}$ $Blm^{tet/tet}$, $Swsap1^{-/-}$ $Blm^{tet/tet}$, or $Spidr^{-/-}$ $Blm^{tet/tet}$ ES cells leads to only a modest reduction in cell number (Fig. 3g and Supplementary Fig. 9e). Thus, the activity of SWS1–SWSAP1–SPIDR not only drives SCEs and LOH in $Blm$ mutant cells, but it also interferes with BLM-dependent HDR and impels the slow growth of these cells.

**Loss of SWS1–SWSAP1–SPIDR prolongs embryonic survival of $Blm$ mutants.** BLM is essential for embryonic development in the mouse[49]. $Blm$ mutant embryos are developmentally delayed, show increased apoptosis and severe anemia, and die by embryonic day (E) 13.5[23]. Given that loss of SWS1–SWSAP1–SPIDR restores cell proliferation in the absence of BLM and drastically reduces the high level of SCEs, we asked whether loss of SWSAP1 might also prolong the development of $Blm^{-/-}$ mice. As expected, $Swsap1^{-/-}$ animals are born at the expected Mendelian ratio (Supplementary Fig. 9f)[7] and viable $Blm^{-/-}$ mice are not obtained. Viable $Swsap1^{-/-}$ $Blm^{-/-}$ mice are also not obtained, indicating that $Swsap1$ mutation cannot fully rescue the survival of $Blm$ mutant mice.

To determine whether SWSAP1 loss could delay the embryonic lethality associated with $Blm$ mutation, we performed timed matings and collected embryos at E15.5. At this stage, $Blm^{-/-}$ mutant embryos are greatly underrepresented from the expected Mendelian ratio (five-fold) (Fig. 4a and Supplementary Fig. 9g), and those few that are obtained are dead, exhibiting developmental delay and severe anemia (Fig. 4b and Supplementary Fig. 10a), as reported. By contrast, $Swsap1^{-/-}$ $Blm^{-/-}$ embryos are obtained at nearly the expected ratio, and while a third of these phenocopy the few $Blm^{-/-}$ embryos that reach E15.5, two thirds are viable at this stage. Although still smaller, these $Swsap1^{-/-}$

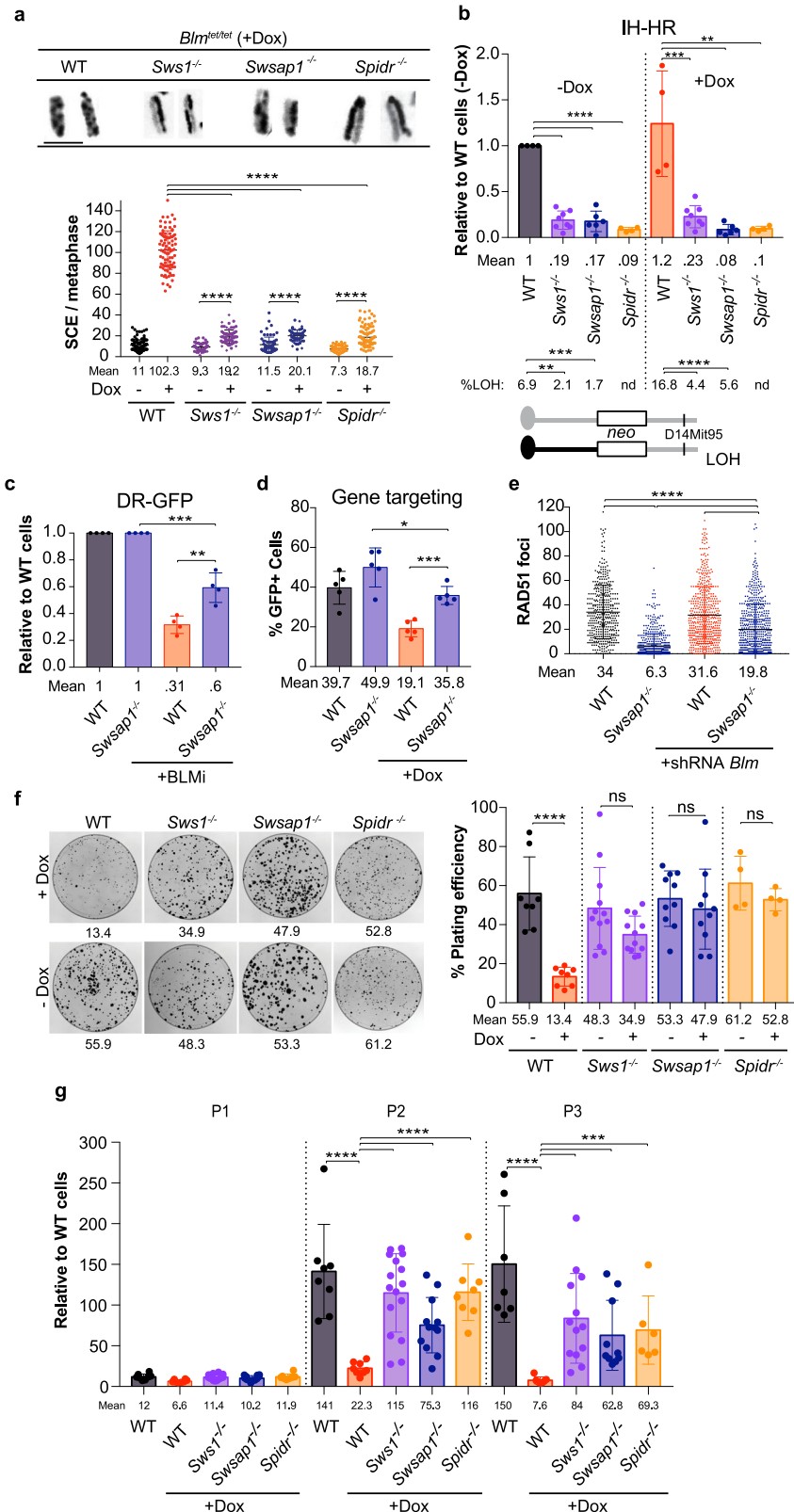

*Blm*$^{-/-}$ embryos are more developed, exhibiting hematopoiesis and fully formed digits. We performed a more limited analysis at E12.5 and found that the *Swsap1*$^{-/-}$ *Blm*$^{-/-}$ embryo obtained at this stage was also more developed than the E12.5 *Blm*$^{-/-}$ embryos (Supplementary Fig. 10c). Thus, loss of SWSAP1 can extend embryonic development in the absence of BLM.

HDR is critical to repair DNA damage arising during neural development, such that in HDR mutant apoptosis of proliferating cells is observed from early to mid-gestation in the brain[50,51]. *Swsap1*$^{-/-}$ embryos have a small increase in apoptosis in the proliferative ventricular zone (VZ) of the forebrain compared to controls at both E12.5 and E15.5 (Fig. 4b, c and Supplementary

**Fig. 3 Mitotic phenotypes associated with BLM depletion are dramatically ameliorated by loss of SWS1–SWSAP1–SPIDR. a**. Loss of SWS1–SWSAP1–SPIDR substantially reduces SCEs in BLM-depleted ES cells. Representative metaphase chromosomes showing SCEs are shown on the left with quantification upon BLM depletion on the right. $Blm^{tet/tet}$ cells that otherwise wild-type or mutated for $Sws1$, $Swsap1$, or $Spidr$ are treated with Dox (1 μM) to deplete BLM. While BLM depletion alone in $Blm^{tet/tet}$ cells increases SCEs ~nine-fold, BLM depletion in $Sws1^{-/-}$ $Blm^{tet/tet}$, $Swsap1^{-/-}$ $Blm^{tet/tet}$, or $Spidr^{-/-}$ $Blm^{tet/tet}$ cells increases SCEs only by ~2 fold. WT-dox $n = 7$, WT + dox $n = 7$, $Sws1^{-/-}$ -dox $n = 5$, $Sws1^{-/-}$ +dox $n = 5$, $Swsap1^{-/-}$ -dox $n = 4$, $Swsap1^{-/-}$ +dox $n = 4$, $Spidr^{-/-}$ -dox $n = 5$, $Spidr^{-/-}$ +dox $n = 5$, where $n$ is the number of independent experiments. (See also Supplementary Fig. 9a). **b** IH-HR in SWS1–SWSAP1–SPIDR mutant ES cells is not further decreased upon BLM depletion, however, LOH of a marker distal to the $neo$ locus (D14Mit95) is reduced in $neo^+$ clones ($p < 0.0001$), presumably due to reduced crossing over. nd, not determined. Relative to WT cells, colony counts expressed relative to wild-type cells transfected with an empty vector within each experiment. WT-dox $n = 4$, WT + dox $n = 4$, $Sws1^{-/-}$ -dox $n = 8$, $Sws1^{-/-}$ +dox $n = 8$, $Swsap1^{-/-}$ -dox $n = 6$, $Swsap1^{-/-}$ +dox $n = 6$, $Spidr^{-/-}$ -dox $n = 4$, $Spidr^{-/-}$ +dox $n = 4$, where $n$ is the number of independent experiments. (See also Supplementary Fig. 9b). **c** BLM inhibition in wild-type ES cells reduces HDR in the DR-GFP reporter, however, HDR is partially restored in $Swsap1^{-/-}$ cells treated with the inhibitor (BLMi: ML216, 50 μM). Relative to WT cells, % GFP + expressed relative to wild-type cells transfected with an empty vector within each experiment. $n = 4$, where $n$ is the number of independent experiments. **d** BLM depletion in $Blm^{tet/tet}$ ES cells reduces DSB-induced gene targeting at the $Hsp90$ locus, however, gene targeting is partially restored in $Swsap1^{-/-}$ $Blm^{tet/tet}$ cells. $n = 5$, where $n$ is the number of independent experiments. **e** RAD51 focus formation in $Swsap1^{-/-}$ primary MEFs is increased when BLM is depleted. Cells were treated with 10 Gy IR. Scale bar, 10 μm. $n = 3$, where $n$ is the number of independent experiments. (See Supplementary Fig. 9c for SCE analysis when BLM is depleted.). **f** Colony formation is reduced by three to four-fold upon BLM depletion; however, it is restored in BLM-depleted $Sws1^{-/-}$, $Swsap1^{-/-}$, and $Spidr^{-/-}$ ES cells. The numbers shown below each plate indicate the plating efficiency (%). $n = 8$, where $n$ is the number of independent experiments. (See also Supplementary Fig. 9d). **g** Population doubling experiments show reduced ES cell numbers with BLM depletion at passages 2 and 3, which can be significantly restored in the absence of SWS1, SWSAP1, and SPIDR. Relative to WT cells, cell counts expressed relative to wild-type cells within each experiment. $n = 6$, where $n$ is the number of independent experiments. (See also Supplementary Fig. 9e) Error bars in **a**–**g**, mean ± s.d. *$P \leq 0.05$; **$P \leq 0.01$; ***$P \leq 0.001$; ****$P \leq 0.0001$; unpaired $t$ test, two-tailed. All source data are provided in the source data file.

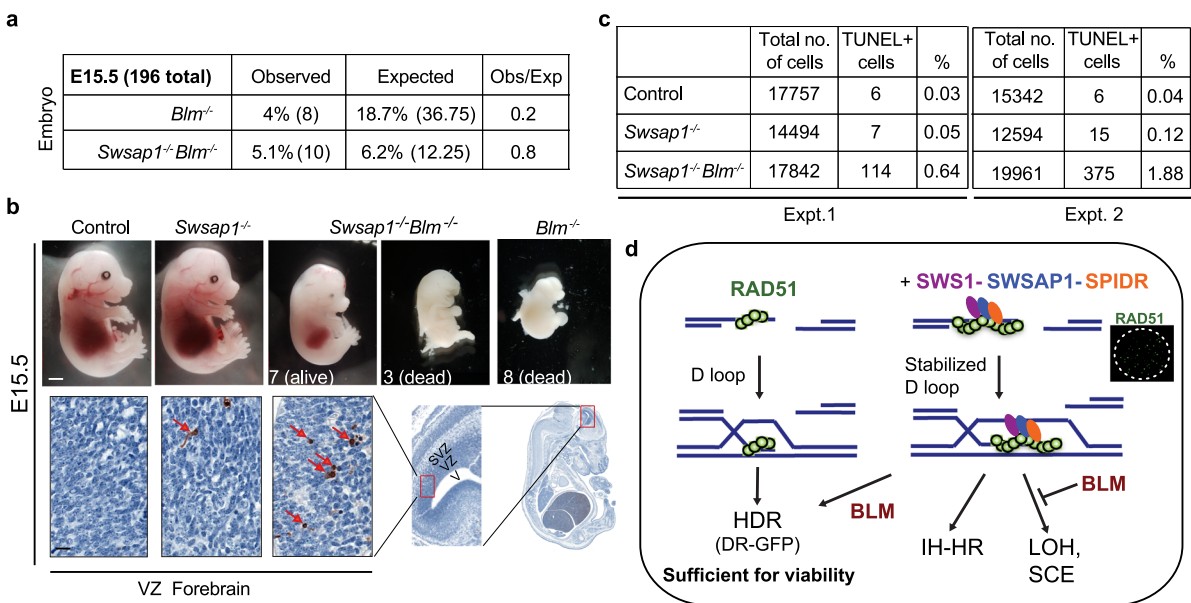

**a**

| E15.5 (196 total) | Observed | Expected | Obs/Exp |
|---|---|---|---|
| $Blm^{-/-}$ | 4% (8) | 18.7% (36.75) | 0.2 |
| $Swsap1^{-/-}Blm^{-/-}$ | 5.1% (10) | 6.2% (12.25) | 0.8 |

**c**

| | Total no. of cells | TUNEL+ cells | % | Total no. of cells | TUNEL+ cells | % |
|---|---|---|---|---|---|---|
| Control | 17757 | 6 | 0.03 | 15342 | 6 | 0.04 |
| $Swsap1^{-/-}$ | 14494 | 7 | 0.05 | 12594 | 15 | 0.12 |
| $Swsap1^{-/-}Blm^{-/-}$ | 17842 | 114 | 0.64 | 19961 | 375 | 1.88 |
| | Expt.1 | | | Expt. 2 | | |

**Fig. 4 Loss of SWSAP1 prolongs embryonic survival of $Blm$ mutants. a** $Blm^{-/-}$ embryos at E15.5 are recovered at five-fold lower than expected from the Mendelian ratio, however, $Swsap1^{-/-}$ $Blm^{-/-}$ embryos are recovered only 20% less often than expected. (See also Supplementary Fig. 9f). **b** Analysis of brains from E15.5 embryos. Although $Swsap1^{-/-}$ embryos have normal embryonic development, the few $Blm^{-/-}$ embryos recovered at this stage are dead. By contrast, most $Swsap1^{-/-}$ $Blm^{-/-}$ are alive, but smaller in size, although a few resemble $Blm^{-/-}$ embryos. Scale bar, 1 mm. Few apoptotic cells are observed in the ventricular zone (VZ) of the forebrain of $Swsap1^{-/-}$ embryos, while $Swsap1^{-/-}$ $Blm^{-/-}$ embryos show numerous TUNEL-positive cells (red arrows). The region analyzed in embryos is progressively depicted (red rectangles), with the ventricle (V), ventricular zone (VZ), and sub-ventricular zone (SVZ) indicated. Scale bar, 50 μm. (See also Supplementary Fig. 10a, b). **c** Quantification of TUNEL positive cells in the VZ from two experiments. **d** Model for the role of the SWS1–SWSAP1–SPIDR complex and its genetic interaction with BLM in multiple HDR outcomes. Right, SWS1–SWSAP1–SPIDR functions to stabilize RAD51 nucleoprotein filaments, which can give rise to visible foci and stabilized D loops. These stabilized intermediates are required for IH-HR and the efficient formation of dHJs. dHJs can be dissolved by BLM or resolved by strand nicking to generate crossovers with associated LOH when homologs are involved or SCEs when sister chromatids are involved. Left, smaller, or less stable RAD51 filaments in the absence of SWS1–SWSAP1–SPIDR are sufficient for intrachromosomal HDR through the synthesis-dependent strand annealing pathway, for example, as assayed in the DR-GFP reporter. In addition to dissolution, BLM is known to unwind D loops to promote synthesis-dependent strand annealing; the requirement for this activity may be less in SWS1–SWSAP1–SPIDR mutant cells since RAD51 nucleoprotein filaments are predicted to be inherently less stable, partially restoring HDR, as assayed with the DR-GFP reporter or by gene targeting.

Fig. 10b, c). By contrast, the viable $Blm^{-/-}$ embryo obtained at E12.5 had numerous apoptotic cells in its malformed forebrain. $Swsap1^{-/-} Blm^{-/-}$ embryos exhibited elevated levels of apoptosis at both stages compared with $Swsap1^{-/-}$ or control embryos, but not as high as the $Blm^{-/-}$ embryo. These results may be explained by the partial, but not total, amelioration of SCE and HDR phenotypes in $Blm$ mutants by concomitant $Swsap1$ mutation.

## Discussion

RAD51 nucleoprotein filaments function at a critical step in HDR, namely the invasion of homologous sequences that can template repair[52,53]. However, factors that direct these RAD51 nucleoprotein filaments to initiate distinct types of HDR have not been identified. Although proteins that nucleate RAD51 nucleoprotein filaments like BRCA2 are apparently critical for all types of HDR, we have uncovered a surprising control of HDR by the SWS1–SWSAP1–SPIDR complex. This complex, which contains the non-canonical RAD51 paralog SWSAP1, promotes certain types of events while having a little discernible impact on others. A distinct role for the complex clarifies why $Sws1$ and $Swsap1$ mutant mice[7], and, as reported here, $Spidr$ mutant mice are viable. Loss of these genes in humans is also likely to be compatible with life but lead to infertility. In line with this, two patients homozygous for a $SPIDR$ truncation allele have been reported to be developmentally normal except with delayed progression to puberty and ovarian dysgenesis[54].

Our study has a broad impact on understanding the role of factors in HDR, whereas also providing specific insight into the role of SWS1–SWSAP1–SPIDR. HDR proficiency is typically evaluated in cells by DNA damage-induced RAD51 focus formation and/or by DSB-induced HDR in reporters that measure intrachromosomal events (DR-GFP). Canonical RAD51 paralog mutants are defective in both assays[22,55–58]. However, we find that RAD51 focus formation is substantially reduced in primary mouse fibroblasts deficient for SWS1, SWSAP1, or SPIDR treated with DNA damaging agents, consistent with previous reports in human cell lines[8,10,13]. This parallels what is observed in $Sws1^{-/-}$, $Swsap1^{-/-}$, and $Spidr^{-/-}$ mouse spermatocytes, in which meiotic RAD51 and DMC1 focus formation are also reduced. By contrast, HDR as assayed between direct repeats (intrachromosomal HDR), which can reflect HDR within a chromatid or between sister chromatids[59], is not reduced in $Sws1^{-/-}$, $Swsap1^{-/-}$, or $Spidr^{-/-}$ mutant mouse fibroblasts or ES cells.

RAD51 filament formation is clearly required for HDR as disrupting filaments through BRC3 expression reduces HDR. However, observable RAD51 focus formation, which involves a large protein assembly and so likely has a low threshold for detecting defects, is apparently not required for all types of HDR. In particular, HDR in reporters like DR-GFP typically involves short sequence repeats (<1 kb) and gives rise to short gene conversion tracts (typically <50 bp)[46], such that only transient or limited RAD51 nucleoprotein filament formation, below the limit of detection, may be required. Given that $Sws1$, $Swsap1$, and $Spidr$ mutant mice are viable, whereas $Brca2$ and canonical RAD51 paralog mutants are not[22], it seems likely that proficient HDR as measured by the DR-GFP reporter reflects physiological intrachromosomal HDR that is required for survival through embryogenesis, whereas observable RAD51 focus formation is not (Fig. 4d).

In contrast to intrachromosomal HDR, SWS1–SWSAP1–SPIDR is important for efficient HDR between homologs, which was completely unanticipated. We previously showed that $Sws1$ and $Swsap1$ mutant mice have greatly reduced meiotic IH-HR[7] resulting in sterility, and here show that $Spidr$ mutant mice are similarly affected. However, it was unexpected that mitotic IH-HR would also need this complex, given that meiotic IH-HR is a highly specialized process involving DSB formation genome-wide during the development of a distinct chromosome structure and, moreover, that other types of mitotic HDR do not require this complex. Although IH-HR is critical for meiotic chromosome segregation, its role in mitotically dividing cells is more limited and is considered to be potentially deleterious because it can give rise to LOH of tumor suppressor genes[29,30].

These results point to the likelihood of a distinct recombination intermediate during mitotic IH-HR that requires SWS1–SWSAP1–SPIDR but that differs from intermediates sufficient for direct-repeat HDR or gene targeting. Because most mitotic IH-HR events are non-crossovers[30], as are events in the DR-GFP reporter[56], these intermediates cannot be limited to crossover-bound outcomes. Our finding that doubling the amount of RAD51 WT partially suppresses the IH-HR defect when SWS1–SWSAP1–SPIDR is deficient is consistent with the complex promoting the stable assembly of longer RAD51 nucleoprotein filaments (Fig. 4d). Some of these intermediates mature into dHJs that are dissolved by BLM[60]; the absence of SWS1–SWSAP1–SPIDR leads to substantially decreased dHJ intermediates and a consequent decrease in LOH events. Consistent with this model, X-shaped recombination intermediates proposed to be dHJs that accumulate in $sgs1$ yeast are substantially reduced by additional mutation of Shu complex components[20].

A role for the SWS1–SWSAP1–SPIDR complex in the formation of stable recombination intermediates, in particular dHJs, is also supported by the analysis of SCEs, either DNA damage-induced or arising in the absence of BLM. Although the effect is relatively mild for MMS and olaparib-induced SCEs (~30%), it is enormous in the absence of BLM (three-fold), such that SCEs remain only two-fold above background levels. Thus, the canonical phenotype of BLM-mutant cells—extremely high SCEs—is largely suppressed by loss of this complex. It is interesting to note that SWS1–SWSAP1 has recently been shown to interact with the cohesion-associated protein PDS5B[13], raising the possibility that the complex specifically affects the use of the sister chromatid. However, a role in determining partner choice during HDR seems unlikely because both IH-HR and SCEs are reduced in the absence of SWS1–SWSAP1–SPIDR. Moreover, mobility of DNA ends is not significantly different in $Swsap1$ mutants, as might be expected if cohesion were impaired. The genetic interactions we observed between SWS1–SWSAP1–SPIDR and BLM are complex, which is perhaps not surprising given that BLM has multiple biochemical activities. In addition to dHJ dissolution[60], BLM can unwind D loops to promote synthesis-dependent strand annealing (Fig. 4d)[61–64], promote DNA end resection[65], and act as an anti-recombinase, stripping RAD51 off of ssDNA[66]. A model regarding the genetic interaction between SWS1–SWSAP1–SPIDR and BLM have to take into account the partial restoration HDR in both the DR-GFP and gene-targeting assays in the absence of BLM. In this case, the inherently less stable RAD51 filaments and D loops formed in the absence of SWS1–SWSAP1–SPIDR would be expected to have a reduced requirement for the unwinding activity of BLM. On the other hand, the partial increase in RAD51 focus formation upon BLM depletion in $Swsap1$-mutant cells is consistent with the loss of the anti-recombinase activity of BLM.

Individuals with Bloom syndrome, unlike mice, survive embryogenesis but have a small stature and marked susceptibility to various types of cancers[67,68]. The suppression of SCEs and increase in HDR upon BLM deletion by loss of SWS1–SWSAP1–SPIDR may account for the enhanced cell proliferation and the prolonged survival of $Blm$-mutant embryos. Our findings raise the possibility that impairing SWS1–SWSAP1–SPIDR function would suppress

symptoms associated with Bloom syndrome. Because *Sws1*, *Swsap1*, and *Spidr* are not essential genes in mice and at-least *SPIDR* appears to be nonessential in humans[54], interfering with their function may be tolerated in a clinical setting to improve patient outcomes.

## Methods

**Plasmids, immunoprecipitation, and western blotting**. cDNAs for human and mouse SWS1 and SWSAP1 (with or without an N-terminal FLAG epitope tag), SPIDR, and DMC1 were synthesized by integrated DNA technology (IDT). To clone *Sws1*, *Swsap1* (wild-type and point mutations), *Spidr* and *Dmc1*, pCAGGS was digested with XhoI and KpnI and cDNAs were cloned into it using infusion HD cloning system (Takara Cat # 639645). The RAD51, RAD51 K133R, and BRC3 expression vectors are in a pCAGGS backbone[32]. RAD51 I287T was similarly cloned into pCAGGS vector[32]. Synthetic DNA open-reading frames coding for human SWSAP1 long and short isoforms and for mouse SWSAP1 with a C-terminal poly-histidine tag were subcloned in-frame into the pMBP-parallel vector[69]. The constructs were verified by DNA sequencing. A different human RAD51 expression plasmid[70] was used for protein purification.

For immunoprecipitation, 4 µg of the indicated expression vectors were transfected into HEK293 cells using lipofectamine 2000 (Fisher scientific Cat # 11668019). Two days after the transfection, cells were spun at $100 \times g$ for 5 min at 4 °C. Cell pellets were lysed in NETN buffer (100 mM NaCl, 20 mM Tris-Cl pH 8.0, 0.5 mM EDTA, 0.5% (v/v) Nonidet P-40 (NP40)) with protease inhibitor cocktail (Roche Cat # 11873580001) by incubating cells on ice for 30 min with gentle flicking every 5 min and then spun at $20,000 \times g$ for 10 min at 4 °C to collect the supernatant. Protein concentration was determined using a Bradford protein assay (Bio-Rad). Then, 1 mg extract was used for each immunoprecipitation assay. All steps were carried out at 4 °C. To pre-clear the extract, 25 µl of pre-washed mouse IgG agarose beads (Sigma Cat # A0919) were incubated with 1 mg of extract for 30 min followed by centrifugation at $15,000 \times g$ for 10 min. The supernatant was transferred to a new Eppendorf containing 25 µl of anti-FLAG M2 beads (Sigma Cat # A2220) and incubated for 2 h. The anti-FLAG M2 beads with the extracts were spun at 20,000 g for 3 min, the supernatant was discarded, and beads were washed three times with phosphate-buffered saline, with Tween 20 (PBST) (50 mM Tris [pH 7.5], 150 mM NaCl, 0.05% Tween 20). After the third wash, beads were resuspended in 2× sodium dodecyl sulphate sample buffer (NEB Cat # B7703), boiled for 10 min, and spun at $10,000 \times g$ for 3 min. The supernatant was transferred to a new Eppendorf and used for western blots.

To perform western blotting, the supernatant was loaded on a precast sodium dodecyl sulphate–polyacrylamide gel electrophoresis, transferred onto nitrocellulose membrane, and blocked with 5% milk in PBST (50 mM Tris [pH 7.5], 150 mM NaCl, 0.05% Tween 20) for 1 h. For immunodetection, the following antibodies were used: anti-SWSAP1 (Cat # PA5-25460) from Thermo; anti-RAD51 (Cat # PC130) from Millipore; anti-DMC1 (Cat # sc-22768) from Santa Cruz Biotechnology; anti-M2-FLAG- HRP (Cat # A8592) from Sigma.

**Fluorescence resonance energy transfer (FRET) assay**. ARPE-19 retinal pigment epithelial cells purchased from ATCC were cultured in Dulbecco's Modified Eagle Medium (DMEM) supplemented with 10% fetal calf serum and antibiotics. Cells were plated on sterile glass 24 mm coverslips in six-well culture plates. Twenty-four hours later, after reaching 60% confluence, cells were transiently transfected with 0.5 or 1 µg of DNA per plasmid, when two or one plasmids were used, respectively. Transfections were performed using Genjet transfection reagent. Twenty-four hours after transfection, FLIM-FRET (Fluorescence Lifetime Imaging Microscopy-FRET) experiments were performed in a 37 °C incubator using a confocal Leica TCS SP8 SMD mounted on a Leica DMI6000 inverted microscope, equipped with a 63X water immersion objective, a white light laser, hybrid detectors, a single molecule detection unit to perform FLIM measurements and a Leica LASX software with FLIM wizard and SymPhoTime64 software. During FLIM measurements, laser intensity was set to the minimal level that produced at most $2 \times 10^6$ counts per second and the repetition rate was set to 20 MHz. Bi-exponential fitting of the fluorescence lifetime data was performed using Sym-PhoTime64 to obtain amplitude-based average lifetimes.

**Protein purification**. Untagged human RAD51 was purified as described[70]. The co-expressed His-SUMO-SWS1–SWSAP1 construct was transformed in Rosetta/pLysS bacterial cells (Novagen Cat # 70956) and induced to express the recombinant protein overnight at 15 °C after addition of isopropyl β- d-1-thiogalactopyranoside (IPTG) (2 L scale). Collected cells were lysed in 50 mL of 1.6 M NaCl, 20 mM Tris pH 8, 1 mM EDTA, 1 mM DTT and 10% (V:V) glycerol supplemented with 1 mM phenylmethylsulfonyl fluoride (PMSF) and protease inhibitor cocktail (Pierce Cat #A32965). The lysate was sonicated and clarified by centrifugation at $20,000 \times g$ for 30 min and the SUMO-SWS1–SWSAP1 recovered in complex by chromatography over 5 ml HisTrap Excel column (GE Healthcare Cat #17-3712-06) pre-equilibrated in 0.8 M NaCl, 20 mM Tris pH 8, 1 mM EDTA, 1 mM DTT, 10 mM imidazole and 10% (V:V) glycerol (NiA buffer) eluted by raising the imidazole concentration to 0.5 M. Then, 2 µg/mL of SUMO protease[71] was added to the eluate and dialyzed using Snakeskin 7MWCO (Thermo Scientific Cat#68700) in 50 mM KCl, 20 mM Tris pH 8, 1 mM EDTA, 1 mM DTT and 10% (V:V) glycerol (SephA buffer) overnight at 4 °C. The SUMO tag and the

SWS1–SWSAP1 complex were separated over 5 ml S column (GE Healthcare Cat #17-5157-01) pre-equilibrated in SephA buffer by raising gradually the salt to 1 M KCl. MBP-SWSAP1-His (mouse, human short and human long) and MBP-SWS1–SWSAP1-His constructs were transformed into Rosetta/pLysS bacterial cells (Novagen Cat # 70956) and induced to express the recombinant protein overnight at 15 °C after addition of IPTG (2 L scale). Collected cells were lysed in 50 mL of 0.5 M NaCl, 20 mM Tris pH 8, 1 mM EDTA, 1 mM DTT and 10% (V:V) glycerol) supplemented with 1 mM PMSF and protease inhibitor cocktail (Pierce Cat #A32965). The lysate was sonicated and clarified by centrifugation at $20,000 \times g$ for 30 min and the fusion protein recovered by chromatography over 5 ml MBPTrap column (GE Healthcare Cat #28918779) adding 20 mM maltose in the elution buffer. The eluate was next passed through a 5 mL HisTrap Excel column (GE Healthcare Cat #17-3712-06) pre-equilibrated in NiA buffer, eluted by increasing the imidazole concentration to 0.5 M. Purified proteins were dialyzed using Sna-keskin 7MWCO (Thermo Scientific Cat#68700) in 150 mM KCl, 20 mM Tris pH 8, 1 mM EDTA, 1 mM DTT, and 10% (V:V) glycerol overnight 4 °C. The preparation was concentrated using Vivaspin 5 K MWCO (GE Healthcare Cat #28-9323-59), aliquoted, and stored at −80 °C.

MBP-BLM (construct pMM1557) and MBP-SPIDR (construct pMM1559) were expressed in HEK293 cells grown in suspension. In brief, cell pellets ($10^8$ cells) were resuspended in 10 mL of buffer T500 (0.5 M NaCl, 20 mM Tris pH 8, 1 mM EDTA, 1 mM DTT and 10% glycerol) supplemented with 1 mM PMSF and protease inhibitor cocktail (Pierce Cat #A32965). After sonication for 30 s, lysates were clarified by centrifugation at $20,000 \times g$ and the fusion proteins purified by amylose resin chromatography, eluted with T500 supplemented with 20 mM maltose, and frozen at −80 °C.

**Protein–protein binding assays**. Binding reactions (100 µL) were assembled by mixing RAD51 and various forms of MBP-SWSAP1 each at 0.4 µM final concentration in buffer B (100 mM KCL, 40 mM Tris pH 7.5, 0.05% (V:V) Triton X-100 and 100 µg/mL acetylated bovine serum albumin (BSA) (Invitrogen Cat #AM2614). Reactions were incubated for 20 min at 37 °C in a rotary shaker (600 rpm). Next, the equivalent of 5 µL of magnetic bead slurry (NEB Cat #E8037S, anti-MBP antibody coupled to superparamagnetic beads) passivated with buffer B was added to each reaction and further incubated for 1 h at 37°C. After the reaction, beads were collected with a magnet and washed twice with 100 µL of buffer B and resuspended in 50 µL of Laemmli buffer. The bead pellets (40%) were fractionated by denaturing SDS–PAGE and the gels stained with Coomassie blue or silver for detection of the proteins.

**RAD51, RPA foci formation, and γH2AX staining**. For RAD51 focus counts and HDR assays in primary cells, the following mouse alleles were used[7]: *Sws1*, *Δ1 A*, and *Swsap1*, *Δ131*. To prepare ear fibroblasts, ear tips were cut from 2 to 3-month old mice in accordance with IACUC guidelines. Ears were minced with a razor blade and dissociated on a 37 °C shaker for 3 h in 3 ml DMEM medium containing 2 mg/ml collagenase A (Roche Cat #11088793001). The dissociated ear fibroblasts were filtered through a 70-µm strainer, and pelleted by centrifugation at $100 \times g$ for 5 min. The pellet was resuspended in DMEM medium and plated in a six-well plate.

Primary ear fibroblasts were expanded and plated in eight-well chamber slides (Thermo Fisher Scientific Cat # C10312) at a density of 20,000 cells per well overnight, followed by 10 Gy IR and 2 h recovery or indicated times of recovery. To label S phase cells, EdU was added right before the IR treatment or 2 h before the cells were fixed onto the slides. For RPA foci staining, cells were pre-extracted with cold 0.5% Triton X-100/PBS on ice for 5 min and washed once with PBS before fixing To fix, cells were incubated in fresh 3% paraformaldehyde/2% sucrose/PBS at room temperature for 10 min and stored at 4 °C. For staining, slides were washed once with PBS, permeabilized in 0.5% Triton X-100/PBS at RT for 10 min, followed by EdU Click-iT (Thermo Fisher Scientific Cat # C10640) to stain S phase cells, and blocked in 1% IgG-free BSA (Jackson ImmunoResearch Cat # 001-000-162) /0.1% Triton X-100/PBS on a shaker at room temperature for 30 min. Slides were then incubated with RAD51 antibody (1:400; Calbiochem Cat # PC130) or RPA2 antibody (1:400; Cell Signaling Cat # 2208) or γH2AX antibody (1:500; Millipore Cat # 05-636) for 2 h in blocking buffer, washed with PBS three times, followed by incubation with an Alexa Fluor 488-conjugated secondary antibody (Life Technologies, Cat # A21206) for 1 h in blocking buffer, and then washed with PBS 3 times. Slides were fixed in ProLong Gold Antifade Mountant with DAPI (Molecular Probes Cat # P36935).

**DR-GFP and IH-HR assays and LOH analysis**. Primary mouse-ear fibroblasts were cultured in DMEM-HG medium supplemented with 10% FBS (Thermo Fisher Scientific Cat # SH30070.03), 1× Pen-Strep, 1× MEM/NEAA, and 1× L-Gln. For DR-GFP assays in mouse ear fibroblasts, $0.1 \times 10^6$ cells were plated a day before in a six-well dish, and the next day they were infected with I-SceI lentivirus[72]. 24 h later, the virus was removed, cells were washed with PBS, and fresh media was added to plates. The percent GFP+ cells were analyzed 48 h later by flow cytometry. Site-loss was determined by PCR amplification of a fragment around the position of the I-SceI site in DR-GFP and then cleaving the fragment in vitro with I-SceI endonuclease (NEB Cat # R0694S); it is calculated as the ratio of the amplified

fragment cleavable by I-SceI to the total amount of amplified fragment (see diagram in Supplementary Fig. 6b)[25].

All mouse ES cell lines were cultured in DMEM-HG medium supplemented with 12.5% stem cell grade FBS (Gemini), 1× Pen-Strep, 1× MEM/NEAA, 1× L-Gln, 833 U/ml LIF (Gemini Cat # 400–495), and 0.1 mM β-mercaptoethanol. Culture dishes were pre-coated with 0.1% gelatin. For DR-GFP assays, $5 \times 10^6$ ES cells were electroporated with 30 µg I-SceI expression plasmid with or without other expression vectors. The trypsinized cells were resuspended in 600 µl Opti-MEM, mixed with plasmids, and pulsed with a Gene Pulser Xcell (Bio-Rad) in a 0.4 cm cuvette at 250 V/950 µF. After pulsing, the cells were mixed with a growth medium and plated to a 6-cm dish. The percent GFP+ cells were analyzed after 48 h by flow cytometry. For experiments with the BLM inhibitor ML216 (Sigma Cat # SML0661), cells were electroporated with I-SceI expression plasmid and resuspended in growth media containing the inhibitor. The inhibitor was replaced after 24 h and flow cytometry was performed after another 24 h.

For IH-HR assays, $15 \times 10^6$ ES cells were electroporated with 30 µg I-SceI expression pCBASce (Addgene 26477) with or without other expression vectors. The trypsinized cells were resuspended in 600 µl PBS, mixed with plasmids, and pulsed with a Gene Pulser Xcell (Bio-Rad) in a 0.4 cm cuvette at 250 V/950 µF. After pulsing, the cells were mixed with 10 ml growth medium and plated to 5–10-cm dishes. Then, 24 h later, one plate was trypsinized to count live cells and the rest of the plates were washed once with PBS and media containing 200 µg/ml G418 (Gemini Cat # 400-111 P) was added to select for *neo+* colonies. G418 media was replaced every 3 to 4 days until the colonies were visible. The plates were fixed in methanol, stained with Giemsa (Life Technologies Cat # 10092-013) and colonies were counted from four plates and the average number of colonies were plotted in graphs. For IH-HR experiments in Fig. 2B, IH-HR is determined by relative fold IH-HR/plating efficiency.

To perform LOH analysis, *neo* + colonies from each condition were picked into a 96-well plate and cultured till confluency. The cell lysate was prepared using lysis buffer (10 mM Tris pH 8.0, 0.45% NP40, 0.45% Tween 20, and proteinase K 50 µg/ml). The amplification of the distal D14Mit95 marker was done using 1 µl of the lysate[30].

**Gene targeting at the *Hsp90* locus**. For gene targeting at the *Hsp90* locus in mouse ES cells, $0.5 \times 10^6$ cells were plated in a six-well dish, and they were transferred to a 10-cm dish the next day. After 48 h, cells were transfected with a plasmid with the targeting vector, a promoterless ZsGreen flanked by homology arms to the *Hsp90* locus, and plasmid DNA for expression of Cas9 and a gRNA directed towards the *Hsp90* locus[26]. Cells were seeded in a six-well dish after transfection and flow cytometry was performed 4 days later for cells expressing ZsGreen. For experiments where BLM expression was repressed, cells were cultured continuously with 1 µM Dox (Sigma Cat # D9891).

**SCE assay**. For SCE assays, $0.4 \times 10^6$ ES cells or MEFs were seeded in 6-cm dish and the next day BrdU (10 µM) was added for 24 h for at least two rounds of DNA replication. For MMS experiments, cells were transiently exposed to MMS (0.5 mM) for 1 h and then were washed with PBS and placed with media containing BrdU. For olaparib treatment, cells were cultured in media containing olaparib (20 nM) and BrdU for ~17 h. For depletion of BLM, cells were continuously cultured in the presence of 1 µM Dox.

Cells were treated 48 h later in media containing colcemid (0.03 µg/ml) for 40 min. Media was collected in a 15-ml tube, and cells were trypsinized and then collected in the same tube. These cells were centrifuged for 5 min at $100 \times g$ and the media was removed. Pre-warmed KCl (75 mM, 4 ml) was added to the pellet and then the tube was inverted gently to resuspend the cells and then incubated for 10 min at 37 °C. Five drops (~100 µl) of cold fixing solution (3:1 v/v methanol-acetic acid) were added to each tube. Tubes were inverted gently and then cells centrifuged for 10 min at $100 \times g$, at room termperature. Media was removed, 5 ml of fixative was added to the pelleted cells; the tube was inverted gently and then incubated on ice for 30 min. Cells were again centrifuged for 10 min at $100 \times g$, at 4 °C. These steps were repeated three times and finally, cells were resuspended in ~500 µl fixative solution. Then, 10 µl of cells were spotted on a slide from a distance of ~10 cm. Aged slides were submerged for 45 min in 50 ml of 0.5× SSC (Saline-sodium citrate) containing Hoechst 33258 (2 µg/ml). Slides were washed for 5 min in 2× SSC and then were immersed in 50 ml 2× SSC and exposed for 1 h to UV light. Next, the slides were incubated in 0.5× SSC at 60°C for 1 h, stained for 15 min in a jar with Giemsa solution in Sorensen phosphate buffer (0.133 M $Na_2HPO_4$, 0.133 M $KH_2PO_4$) at room temperature. The slides were dried at room temperature and then the coverslip was added to image them on a Delta vision ultrahigh-resolution microscope using a 100X oil-immersion objective.

**Clonogenic survival and population doubling assays**. For clonogenic survival assays, 500 and 1000 cells from IH-HR experiments were plated in a 10-cm plate and were allowed to grow for a week. Plates were then fixed in methanol, stained with Giemsa, and colonies were counted; percent plating efficiency was calculated as the number of colonies divided by the number of plated cells. For population doubling assays, 50,000 cells were plated in 12-well plates at passage 0, and cells

were counted every two days until passage 3. For Dox experiments, cells were precultured continuously for 48 h in 1 µM Dox (Sigma Cat # D9891) and during the population doubling assays.

**Live-cell imaging and analysis for tracking the movement of GFP-tagged RPA32**. Control ($Brca2^{+/-}$ $Swsap1^{+/-}$) and $Swsap1$ mutant ($Brca2^{+/-}$ $Swsap1^{-/-}$) transformed MEFs were cultured in high-glucose DMEM with L-glutamine, 10% fetal bovine serum, and 1% penicillin–streptomycin. MEFs were plated on 35-mm glass-bottom microwell dishes (MatTek Cat #P35GC-1.5-10-C) and transfected with the pEGFP–NLS–RPA32 construct ("RPA-GFP", gift from Jiri Lukas) using Lipofectamine 2000 (Invitrogen, 11668). Then, 10 h following transfection, cells were damaged with 0.5 µg/mL NCS (Sigma-Aldrich Cat #N9162) for 60 min at 37 °C. Cells were washed twice with PBS and allowed to recover for 14 h prior to imaging. Following recovery, cells were imaged on an A1RMP confocal microscope (Nikon Instruments), on a TiE Eclipse stand equipped with a ×60/1.49 Apo-TIRF oil-immersion objective lens, an automated XY stage, and a stage-mounted piezoelectric focus drive. Live-cell imaging was performed in a heated, humified chamber with 5% atmospheric $CO_2$. RFP-GFP foci were imaged using a 488 nm excitation laser and GFP emission filters. Z-series with 0.4 µm intervals were collected throughout selected nuclei every 5 min for 100 min. The focus was maintained by the Perfect Focus System (Nikon).

DSB foci movement was analyzed as follows[31]. In brief, data files exported from Nikon NIS Elements were analyzed in ImageJ[73], where maximum-intensity z-stack projections were generated. The StackReg plug-in was used to correct for cell movements during the duration of imaging and any nuclei with gross nuclear deformations were discarded[74]. Single-particle tracking was performed using the ImageJ Trackmate plug-in[75], and trajectories were subsequently analyzed in MATLAB using the @msdanalyzer[76].

**Yeast-two-hybrid**. For yeast-two-hybrid analysis, pGAD-SWS1 and pGBD-SWSAP1 short or long isoform plasmids were transformed into the *Saccharomyces cerevisiae* strain PJ69-4A and transformed yeast selected for on SC-LEU-TRP medium[13]. Single transformants were grown overnight in SC-LEU-TRP liquid medium and 5 µl of 0.5 $OD_{600}$ were spotted onto SC-LEU-TRP, SC-LEU-TRP-HIS, SC-LEU-TRP-HIS + 3AT, or SC-LEU-TRP-ADE plates. Plates were grown for 2 days at 30°C before being photographed. Images were adjusted identically for brightness and contrast using Adobe Photoshop.

**Generation of *Sws1*, *Swsap1*, and *Spidr* mutant cell lines and *Spidr* mice**. A dual gRNA approach was used to knockout *Sws1*, *Swsap1*, and *Spidr* in the following ES cell lines: J1-DR1[77] and 129/B6 $Blm^{tet/tet}$[42]. The gRNA sequences were cloned into the dual Cas9/gRNA expression vector pSpCas9(BB)-2A-Puro (PX459, Addgene 48139) according to published protocols[78]. Multiple colonies were picked. Those that gave PCR products indicating a deletion between the two gRNAs were sequenced to identify out-of-frame mutations. In addition to clones that had lost the wild-type alleles, two clones were isolated which maintained the wild-type sequence as additional controls for the DR-GFP assays.

*Sws1* genotyping was done using the following PCR primers: *Sws1*-A: 5′-CCTGCAGGGGCGCGTGAAGTTC-3′, *Sws1*-B: 5′-ACCGGCTCGCACTCAGGGATC-3′ under the following conditions: 94°C, 3 min; 35 cycles of 94°C, 30 s; 55°C, 1 min and 65°C, 30 s; and a final extension of 72°C, 5 min[7].

*Swsap1* genotyping was done using the following PCR primers *Swsap1*-C: 5′-TCTGTGAACTATAGCCAATGAGGC-3′, and *Swsap1*-D: 5′-AACTGTCACTCAGGCGCGAACTAG-3′ under the following PCR conditions: 94°C, 3 min; 35 cycles of 94°C, 30 s; 55°C, 1 min and 65°C, 30 s; and a final extension of 72°C, 5 min[7].

*Spidr1* genotyping was done using the following PCR primers *Spidr*-F: 5′-CCATGTCAAGTTTCCGAGTCATTC-3′, and *Spidr*-R: 5′-AGCATCCTTAGTATGCATAGATTCTAC-3′ under the following conditions: 95°C, 3 min; 35 cycles of 95°C, 30 s; 60°C, 30 s and 68°C, 1 min; and a final extension of 68°C, 5 min. PCR products were run on a 2.4% agarose gel to resolve the wild-type band from the mutant band. The wild-type product is 470 bp. To generate *Swsap1 Spidr* double mutants, *Spidr* was knocked out using clone *Swsap1#69* in J1-DR1 and *Swsap1#39* in 129/B6 $Blm^{tet/tet}$ ES cells.

To generate *Spidr* knockout mice, dual gRNA approach was used as described above for ES cells. C57BL6/J zygotes were electroporated (Bio-Rad GenePulser Xcell) by 7 pulses (30 V–3 ms pulse, 100 ms interval) with a mix containing 100 ng/µl HiFi Cas9 nuclease (IDT) and 50 ng/µl of each sgRNA (Millipore-Sigma) for *Spidr*. Electroporated zygotes were transferred the same day into pseudopregnant recipients. In all, 68 founders were born and after genotyping them, a heterozygous mouse with a Δ77 allele was chosen for further analysis.

**Timed matings and embryo analysis**. Genotyping of *Blm* mice uses the primers *Blm* F 5′-CACTGAGGAATGTTTACCCACCACC-3′; *Blm* neo, 5′-GCAGCCTCTGTTCCACATACACTTC-3′;*Blm* R, 5′-CCCAGTCATCATCTTCATCATCATC-3′ and the following conditions: denaturation at 95°C for 5 min was followed by 35 cycles of 94°C for 0.5 min, 60°C for 1 min, and 72°C for 2 min[23]. The products are as follows: wild-type allele, 320 bp; mutant allele: 190 bp. $Swsap1^{+/-}$ $Blm^{+/-}$ males were mated with $Swsap1^{+/-}$ $Blm^{+/-}$ females and plugs were checked to determine a pregnancy start date. Pregnant females were sacrificed on 12th or 15th day.

Uterine horns were dissected, embryos were collected, photographed, and then fixed overnight in 4% paraformaldehyde (PFA). Embryos that had a heartbeat were marked as alive in the analysis. To perform TUNEL analysis, embryos were fixed in 4% PFA and stained with hematoxylin and TUNEL (Roche, 03333566001 and 11093070910, respectively).

**Histology.** Testes were collected from animals between 6 and 8 weeks of age, fixed in 4% PFA, and sectioned and stained with Hematoxylin and Eosin (H&E). Staging of seminiferous tubules[79] was done on the H&E-stained testes sections.

**Spermatocyte chromosome spreads and immunofluorescence.** Spermatocytes for surface spreading were prepared from testes of ~2-month-old mice using established methods[80]. The following primary antibodies were used in dilution buffer (0.2% BSA, 0.2% fish gelatin, 0.05% Triton X-100, 1× PBS), with incubation overnight at 4℃: goat anti-SYCP3 (Santa Cruz Biotechnology Cat# sc-20845; 1:200), rabbit anti-RAD51 (Calbiochem Cat# PC130; 1:200), rabbit anti-DMC1 (Santa Cruz Biotechnology Cat# sc-22768; 1:200). This was followed by incubation with the following secondary antibodies at 1:500 dilution for 1 h at 37℃: 488 donkey anti-rabbit (Life Technologies Cat#A21206), donkey 594 anti-goat (Invitrogen Cat# A11058). Coverslips were mounted with ProLong Gold antifade reagent with DAPI. Immunolabeled chromosome spread nuclei were imaged on a Deltavision microscope using ×100 oil-immersion objective. Only foci colocalizing with the chromosome axis were counted.

**Animal work.** The care and use of mice in this study were performed in accordance with a protocol approved by the Institutional Animal Care and Use Committee (IACUC) at Memorial Sloan Kettering Cancer Center (MSK). Mice were housed under Federal regulations and policies governed by the Animal Welfare Act (AWA) and the Health Research Extension Act of 1985 in the Research Animal Resource Center (RARC) at MSKCC, and was overseen by IACUC.

**Statistics and reproducibility.** Statistical analyses were performed using a two-tailed, unpaired $t$ test in GraphPad Prism 7. Error bars, mean ± s.d.; ns, not significant; $*P \le 0.05$; $**P \le 0.01$; $***P \le 0.001$; $****P \le 0.0001$. Exact $P$ values are provided in the Supplementary Data File. The number of experiments repeated independently is provided in the figure legend.

**Reporting summary.** Further information on research design is available in the Nature Research Reporting Summary linked to this article.

## Data availability

Source data are provided with this paper in the Source Data File or are available from the correspondent author upon request. (m-jasin@ski.mskcc.org).

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

## Acknowledgements

We thank members of the Jasin laboratory for helpful discussions, especially Tai-Yuan Yu, Agnieszka Lukaszewicz, and Yufuko Akamatsu, as well as Katia Manova and members of the MSK Molecular Cytology core facility for technical help, Agnel Sfeir for the *Hsp90* targeting plasmids, and Peter Romanienko for generating *Spidr* knockout mice. Core facilities at MSK are supported by a Cancer Center Support Grant (NIH P30 CA008748). The research was supported by NIH F32 GM110978 (R.P.), a Paoli-Calmettes Institute Ph.D. fellowship (F.M.), NIH F31 ES027321 (M.R.S.), NIH R01 ES030335 and ACS Research Scholar Grant 129182-RSG-16-043-01-DMC (K.A.B.), the French National League against Cancer (M.M.), and the MSK Functional Genomics Initiative, Cycle for Survival, NIH R35 GM118175, R01 CA185660, and R35 CA253174 (M.J.).

## Author contributions

R.P., F.V., and M.J. conceived the project and designed experiments. R.P., T.S., B.T., R.W., P.X.L., T.W., and F.V. performed experiments, except in vitro protein interactions (F.M.), FRET (E.C.B.D., P.M.K.), 129/B6 *Sws1* and *Swsap1* disruption (M.R.S.), Y2H (H.L.R.), and chromosome mobility (J.A.Z.). R.P., F.V., K.A.B., J.G., M.M., and M.J. provided supervision. R.P., F.V., and M.J wrote the manuscript.

## Competing interests

The authors declare no competing interests.
