## [Peer Review File · Nature Communications]

REVIEWER COMMENTS

Reviewer #1 (Remarks to the Author):

Review on Prakash et al.
Ncomm-20-19537-T

The paper by Prakash et al describes the characterization of a newly identified complex, SWS1-SWSAP1-SPIDR, which is involved in homology-directed repair (HDR) in mouse cells and mouse. SPIDR, which also interacts with Blm helicase, is a novel component of the complex, whose have not been characterized well. The authors showed that mouse cells knocked out for Sws1, Swsap1 and Spidr are proficient in some recombination such as intra-chromosomal recombination and gene targeting, but are defective in interhomolog recombination in both mitosis and meiosis and in sister chromatid exchange, supporting an idea that the three proteins form a complex for a unique pathway for HDR. Indeed, all three mutants are deficient in RAD51 focus formation in somatic cells and spermatogonia. Interestingly, the deletion of Sws1, Swsap1 and Spidr partially rescues cellular lethality conferred by Blm depletion. The paper contains various cellular analyses of recombination and DNA repair as well as analysis of mouse (embryo and testis) and provide a new insight on the molecular function of the SWS1-SWSAP1-SPIDR. The experiments have been technically conducted well and results are convincing with large quantity. It seems worthwhile publishing in Nature Communications. Because of large amounts of the data in the paper, it is very helpful for readers to explain more detail in some results in the paper. With this concern, there are several minor points described below should be addressed.

Major point:

As the authors have known, BLM can interact with SPIDR in human cells (Wan et al 2013). It would be nice to confirm this and to check the binding of Spidr to Blm in mouse cells and the binding of Swsap1 (or Sws1) with Blm. It will be easily possible since the authors have shown the interaction of Flag-SPIDR with SWSAP1 and RAD51 (Supplementary Figure 1g, j). This would be important to clarify whether BLM is a component of SWS1-SWSAP1-SPIDR or BLM forms a distinct complex with SPIDR.

Minor points:

1. Figure 1A; this presentation is misleading since there are few biochemical evidences to claim SWS1-SWSAP1-SPIDR complex shows 1:1:1 stoichiometry.
2. Figure 1j; It would be nice if staining of RPA (or gammaH2AX) is included as a control.
3. Page 5, last three lines and Figure 1i; This sentence and associated Figure are not necessary here. It is very much speculative to conclude here that SWS1-SWSAP1-SPIDR fills a gap between RAD51 filaments. Please rewrite here to reflect the results in the paper and remove it.
4. Page 6, third paragraph, line 2; In Supplemental Figure 2c and 4b, the authors used ES cells with Blm tet/tet rather than WT. This does not affect the interhomolog (IH) recombination frequencies?? The authors need more words why this allele is used for the experiment.
5. Figure 3C; In the RPA-FLAP assay, please explain a graph in bottom for distance measurement in more detail.
6. Figure 3d and S6b; If the value is "relative ratio to WT", WT frequency should be included even it shows "1.0".
7. Figure S1i; Does Spidr interact with Dmc1?
8. Figure 1g; gene targeting assay should be illustrated in Supplement, particularly it is important to show a construct the plasmid with ZsGreen for the transfection is linearized or not and what kind of homology and heterology it bears to the target.
9. Figure 2a; It would be helpful if the authors change the relative color of I-SceI, PacI and NcoI sites in the Figures since NcoI site on P allele seems to be a mutation site with white color (but not on Neo+ allele).
10. Figure S6a; a gel is too dark to see RAD51 band.
11. Figure S6f; "Swsap1" in the right could be mouse Swsap1. If so, the caption should be changed to clarify.
12. Figure 3e and S6g; Although different Y axes are used, the results could come from the same experiment. However, why P-values are different between the two graphs.
13. Page 8, second paragraph and Figure 3g; BRC3 reduced IH-HR in swsap1-/- cells. If so, please

show the P-value in the graph.

14. Page 9, second paragraph: in this LOH assay in the bottom of Figure 3b, ratios (percent) are shown. It would be nice to show actual frequencies of LOH in the absence and presence of Tet and to check statistical differences.

Reviewer #2 (Remarks to the Author):

Prakash et al. provide interesting and valuable new information regarding the role of the SWS1-SWSAP1-SPIDR complex in homology-directed repair (HDR) and reveal a novel function of this complex in promoting inter-homolog homologous recombination (IH-HR). They show that SWS1, SWSAP1, and SPIDR are required for meiosis and RAD51 and DMC1 focus formation in spermatocytes. In addition, they observe a reduced level of RAD51 focus formation upon IR irradiation, but cells deficient for SWS1-SWSAP1-SPIDR perform gene conversion type of HR in the context of the DR-GFP assay at normal levels. Interestingly, the absence of any component of this complex results in decreased IH-HR as measured using the S/P reporter. These results are in contrast to more general disruptions of HDR by the RAD51-KR mutant or a BRCA2 peptide (i.e., both decreased signal in the DR-GFP assay or survival using the S/P reporter). Interestingly, the role of SWS1-SWSAP1-SPIDR in IH-HR appears to be coupled to BLM function. Depletion or inhibition of BLM results in the previously reported elevated sister chromatid exchange (SCE) phenotype, but co-depletion of SWS1-SWSAP1-SPIDR and BLM leads to WT levels of SCEs. The authors propose a model in which the SWS1-SWSAP1-SPIDR complex stabilizes recombination intermediates that are resolved by BLM. In the absence of BLM activity (such as in Bloom syndrome), SWS1-SWSAP1-SPIDR promote SCEs and crossovers because BLM is no longer available to regulate these activities.

The manuscript is well written and the results are impactful. Some small changes are recommended.

1. Figure 1A and Supp. Fig 1 do not add much and implies SPIDR interacts with both SWS1 and SWSAP1 despite the experimental data only showing SWSAP1-SPIDR (Supp. Fig 1G).
2. Would the depletion of SWSAP1 result in a mild hyper-recombination phenotype based on the results in Fig. 1F?
3. Proficient HR despite reduced RAD51 foci after SWS1, SWSAP1, or SPIDR disruption needs to be explained better.
4. Liu et al (PMID:21965664) showed that formation of the SWS1-SWSAP1 complex is required for stability of the partner. Can the authors inform us about the levels of SWS1, SWSAP1, and SPIDR in the various cell lines?
5. In conjunction with the IH-HR result in Supp. Fig 6, does mutation of the SWSAP1 Walker A or B box affect RAD51 focus formation or gene conversion in the DR-GFP assay?

Reviewer #3 (Remarks to the Author):

Summary

In this manuscript, Prakash et al. describe the importance of the SWS1-SWSAP1-SPIDR complex in the control of distinct modes of homologous recombination. This complex is known to play a role in the control of inter-homolog recombination during meiosis. The authors find that this triple S complex also regulates inter-homolog recombination during mitotic HDR, as measured by a genetic assay in human cells. Intra-chromosomal recombination as measured by the DR-GFP assay is largely unaffected. Strikingly, the high rate of SCEs in BLM^{-/-} cells that gives rise to the harlequin chromosome phenotype is rescued upon depletion of components of the SSS complex. The authors propose a model whereby the SSS complex stabilizes RAD51 filaments.

Overall, the experiments presented in the manuscript are well-designed and the data is of high quality. The authors have an abundance of genetic evidence for the role of the SSS complex in regulating inter-homolog recombination in mitosis; additionally, the SCE phenotype in BLM cells is interesting and of potential therapeutic interest. The novelty of this work is high due to the fact

that ablation of any of the SSS complex components largely impairs inter-homolog recombination but not intra-homolog recombination, which is a novel finding in the context of mitotic cells. This “separation of function”-like phenotype is not possible by ablation of RAD51, which impairs recombination generally. However, despite the exciting findings, this work does not provide much mechanistic insights into the role of SWS1-SWSAP1-SPRN complex in IH-HR recombination, which at this stage remains somewhat speculative. A major role for the SWS1-SWSAP1-SPRN complex in regulating RAD51 foci formation during meiosis (IH-HR) has already been established by the same group in a previous work (Abreu et al. 2018, Nature Communications). The lack of a mechanism connecting the SWS1-SWSAP1-SPRN complex and RAD51 dynamics during IH-HR is an important gap in the paper that somewhat reduces its impact and significance. For this reason, the authors should try to better characterize the role and mechanism by which the SWS1-SWSAP1-SPRN complex stabilizes the RAD51 nucleofilament, particularly in the context of BLM deficient cells. The employment of in vitro assays to show the effect of the SSS complex on RAD51 filament stability could be useful (Wang et al. 2015, Molecular Cell: doi: 10.1016/j.molcel.2015.07.009), although that may be beyond the scope of this publication.

Other Major Concerns

- 1- It was confusing whether the authors consider SCEs as intra-chromosomal HDR events or not. SCEs are certainly not inter-homolog HDR events, but whether they are intra-chromosomal or not may be a matter of semantics, or if someone considers chromatids on a replicating chromosome as part of the same chromosome. The authors should clarify their view on this, especially since they do observe a reduction in SCEs upon SSS loss of function. Therefore SCEs are not really behaving as intra-chromosomal HDRs as measured by DR-GFP, but seem to behave as inter-homolog HDR events.
- 2- It would be useful and informative that the authors test if depletion of BLM rescues the Rad51 foci reduction in SSS complex depleted cells. This would better characterize the mutually antagonistic effects of BLM and SSS proteins. Moreover, in the discussion the authors speculate that X-shaped recombination intermediates are elevated in BLM cells, based on evidence from yeast. According to the yeast model, this phenotype would be “rescuable” by depletion of members of the SSS complex. The authors could test this.
- 3- Genetic ablation is used to great effect in this study, but the authors should also test if ectopic overexpression of the SSS complex produces a higher number of SCEs in cells treated with MMS or PARPi. The same should be done for BLM^{-/-} cell lines.
- 4- In Figure 1F and 1G the authors first show no reduction in classical HDR efficiency as measured by DR-GFP in SWS1^{-/-}, SWSAP1^{-/-} and SPDR^{-/-} cells and then, in Figure 1J, they show a great reduction in RAD51 foci formation after IR. This is an intriguing and relevant information to the field. The authors do provide some explanation to this disconnect in the discussion, but it would be useful to go a bit further. Is it just that less stable RAD51 filaments contain less RAD51 molecules and therefore provide lower signal for detection?
- 5- The authors should more rigorously test the possibility that disruption of SSS complex components alters resection dynamics. Currently their only evidence to the contrary is the data presented in figure 2C, which considers only one SSS complex component in the presence of a single drug, NCS.
- 6- Authors should include analysis of gene targeting and RAD51 foci also in SWS1^{-/-} and SPDR^{-/-} cells (Figure 1G, K). Same for figure 2D, E and F.
- 7- I felt that the introduction lacks a better background on SSS proteins. A more in-depth description of their domain organization and some of the previously defined activities of these proteins would help better situate the reader who is unfamiliar with the topic.
- 8- The use of BRC3 peptide and the RAD51 KR mutant is confusing. In the case of RAD51-KR mutant, if it indeed leads to hyper-stabilized RAD51-filaments, why doesn't it improve inter-homolog HR? This should be the case especially in SWS1-depleted cells. Also, my expectation is that the BRC3 peptide should exacerbate the phenotypes of SWS1-depleted cells. But this is not the case, and the authors should clarify.

Minor Concerns

- 1-Figure 1L is confusing and the connection between RAD51 filaments, viability, and fertility is unclear. The authors should simplify the schematic in the form of a flowchart akin to what is described in the figure legend or omit the figure entirely.
- 2- Figure S1F is misleading. Is this a CoIP? Which is the protein immunoprecipitated? Which bands

corresponds to SWS1 and which corresponds to SWSAP1? Where is Flag-SWS1 in the input sample? Same for Figure S1G. Which is the protein immunoprecipitated? For consistency, please show molecular weight of the indicated proteins.

3- Figure S1H: Where are the Flag-SWS1 and FLAG-SWAP1 western blot signals? It looks like SWSAP1 interacts more strongly with RAD51 than SWS1 but is harder to say without an IP control. Authors need to show all the IP controls.

4- In many cases, the information in figures needs to be improved. For example, it is often not clear what is being measured and how. For example, the Y axis title for the bar chart in Figure 2B reads "relative to WT." It should read something along the lines of "IH-HR efficiency (relative to WT)" to indicate more clearly what assay/measurement is being used. Conversely, many of the figure legends are too verbose and include interpretations of the data that is best left for the results/discussion section.

Summary of new data in the following figure panels:

Figure 1a. Purified SWS1, SWSAP1 and SPIDR form a complex.

Figure 1f. *Sws1*^{-/-}, *Swsap1*^{-/-}, and *Spidr*^{-/-} ES cells are proficient at DSB-induced gene targeting.

Figure 2f. Constitutive overexpression of RAD51 WT increases IH-HR in wild-type and *Swsap1*^{-/-} cells, while RAD51 I287T expression reduces IH-HR.

Figure 3b. IH-HR in SWS1-SWSAP1-SPIDR mutant cells is not further decreased upon BLM depletion, however, in *neo*⁺ clones LOH of a marker distal to the *neo* locus (D14Mit95) is reduced (p<0.0001), presumably due to reduced crossing over. (New data is on LOH analysis).

Figure 3e. RAD51 focus formation in *Swsap1*^{-/-} primary MEFs is increased when BLM is depleted.

Figure 4d. (modified) Model for the role of the SWS1-SWSAP1-SPIDR complex and its genetic interaction with BLM in multiple HDR outcomes.

Supplementary Fig. 2a. SWS1-SWSAP1 interaction with BLM.

Supplementary Fig. 2c,e. (modified) Interactions are observed for mouse SWSAP1 with SWS1 (c) and with SPIDR (d) and also between the three proteins with RAD51 (e,f) in co-immunoprecipitation experiments from HEK293 cells.

Supplementary Fig. 2g,h. RAD51 interactions are also observed with human SWS1-SWSAP1 (g) and SWS1-SWSAP1-SPIDR (h) using purified proteins.

Supplementary Fig. 2i,j. Interactions are observed for mouse SWS1-SWSAP1 (i) and SPIDR (j) with DMC1 in co-immunoprecipitation experiments from HEK293T cells.

Supplementary Fig. 6a. (left panel modified, right panel added) Left, HDR results for individual clones using the DR-GFP reporter. Expression of SWSAP1 does not alter HDR in *Swsap1*^{-/-} mutant cells.

Supplementary Fig. 6f,g. Mutant and control primary ear fibroblasts have similar levels of DNA damage upon IR exposure (f) and similar number of RPA foci after treatment with IR.

Supplementary Fig. 7b,c,d,e, f. IH-HR analysis in *Sws1*^{-/-}, *Swsap1*^{-/-}, and *Spidr*^{-/-} ES cells (multiple panels).

Supplementary Fig. 8g. Analysis of mouse and human SWSAP1 Walker A and B mutants in the DR-GFP assay.

Supplementary Fig. 9b. LOH of the distal marker D14Mit95 is reduced in cells mutated for *Sws1* or *Swsap1*.

Supplementary Fig. 9c. Knockdown of BLM in wild-type MEFs leads to an increase in SCE formation, however, *Swsap1*^{-/-} MEFs depleted for BLM show reduced SCE.

Summary of former panels moved to the supplement or deleted:

To Supplement: Figure 1b, Figure 2d

Deleted: Figure 1l, Figure 6g (now in raw data file)

Reviewers' comments in black; authors' rebuttal in blue.

Reviewer #1 (Remarks to the Author):

Review on Prakash et al.

Ncomm-20-19537-T

The paper by Prakash et al describes the characterization of a newly identified complex, SWS1-SWSAP1-SPIDR, which is involved in homology-directed repair (HDR) in mouse cells and mouse. SPIDR, which also interacts with BLM helicase, is a novel component of the complex, whose have not been characterized well. The

authors showed that mouse cells knocked out for *Sws1*, *Swsap1* and *Spidr* are proficient in some recombination such as intra-chromosomal recombination and gene targeting, but are defective in interhomolog recombination in both mitosis and meiosis and in sister chromatid exchange, supporting an idea that the three proteins form a complex for a unique pathway for HDR. Indeed, all three mutants are deficient in RAD51 focus formation in somatic cells and spermatogonia. Interestingly, the deletion of *Sws1*, *Swsap1* and *Spidr* partially rescues cellular lethality conferred by *Blm* depletion. The paper contains various cellular analyses of recombination and DNA repair as well as

analysis of mouse (embryo and testis) and provide a new insight on the molecular function of the SWS1-SWSAP1-SPIDR. The experiments have been technically conducted well and results are convincing with large quantity. It seems worthwhile publishing in Nature Communications. Because of large amounts of the data in the paper, it is very helpful for readers to explain more detail in some results in the paper. With this concern, there are several minor points described below should be addressed.

We appreciate these very positive comments about the manuscript. As requested, more detail is presented for some of the results to assist the reader.

Major point:

As the authors have known, BLM can interact with SPIDR in human cells (Wan et al 2013). It would be nice to confirm this and to check the binding of *Spidr* to *Blm* in mouse cells and the binding of *Swsap1* (or *Sws1*) with *Blm*. It will be easily possible since the authors have shown the interaction of Flag-SPIDR with SWSAP1 and RAD51 (Supplementary Figure 1g, j). This would be important to clarify whether BLM is a component of SWS1-SWSAP1-SPIDR or BLM forms a distinct complex with SPIDR.

Indeed, using purified proteins we were able to detect an interaction between SWS1-SWSAP1 and BLM. We have now included this data as Supplementary Fig. 2a, indicating a broader relationship of the complex members with BLM, which is consistent with similar genetic interactions between each complex member and BLM.

We also attempted to recapitulate the published protein-protein interaction between SPIDR and BLM but were unable to, likely due to technical reasons.

Minor points:

1. Figure 1A; this presentation is misleading since there are few biochemical evidences to claim SWS1-SWSAP1-SPIDR complex shows 1:1:1 stoichiometry.

While it is true that we cannot claim that SWS1-SWSAP1-SPIDR has a 1:1:1 stoichiometry, they form a complex in in-vitro experiments with purified proteins (Fig 1a). We indicated in the figure legend that the top panel is a schematic and provides the color code for subsequent figures.

2. Figure 1j; It would be nice if staining of RPA (or γ H2AX) is included as a control.

We have now included both RPA and γ H2AX data in the supplementary figures, showing that there is no significant difference between control and all three mutants for RPA foci (Supplementary Fig. 6h), indicating there is not a resection defect, and control and *Swsap1* mutant for γ H2AX staining (Supplementary Fig. 6g), indicating that DNA damage is similar.

3. Page 5, last three lines and Figure 1i; This sentence and associated Figure are not necessary here. It is very much speculative to conclude here that SWS1-SWSAP1-SPIDR fills a gap between RAD51 filaments. Please rewrite here to reflect the results in the paper and remove it.

We did not intend to convey that the complex fills a gap but in any case, have removed these sentences and Figure 1i.

4. Page 6, third paragraph, line 2; In Supplemental Figure 2c and 4b, the authors used *Blm* tet/tet rather than WT. This does not affect the interhomolog (IH) recombination frequencies?? The authors need more words why this allele is used for the experiment.

Because we were interested in studying the function of SSS complex and its relationship with BLM in IH-HR, we used *Blm*^{tet/tet} cells in which Dox addition leads to depletion of BLM. However, in the absence of Dox, cells express BLM. To clarify, we have now added the following to the legend of Supplementary Fig. 3c:

“129/B6 Blm^{tet/tet} were used in c to be able to regulate BLM expression (-Dox, BLM is expressed; +Dox, BLM is not expressed).”

And we added the following sentence to the results:

*“The 129/B6 ES cells contain modified *Blm* alleles (*Blm*^{tet/tet}), such that doxycycline (Dox) addition results in depletion of BLM³⁹.”*

5. Figure 3C; In the RPA-FLAP assay, please explain a graph in bottom for distance measurement in more detail.

We expanded the Fig. 2c legend and also added more text in the results section to clarify.

6. Figure 3d (should be 2d?) and S6b; If the value is “relative ratio to WT”, WT frequency should be included even it shows “1.0”.

The value is actually relative to complementation by mouse SWSAP1. We have now specifically indicated this on the graph (now as Fig. S8b).

7. Figure S1i; Does Spidr interact with Dmc1?

As requested, we performed SPIDR and DMC1 co-immunoprecipitation experiments in 293T cells and found that SPIDR interacts with DMC1 (Fig S2i).

8. Figure 1g; gene targeting assay should be illustrated in Supplement, particularly it is important to show a construct the plasmid with ZsGreen for the transfection is linearized or not and what kind of homology and heterology it bears to the target.

This assay has been described before by another lab, but we have now included a diagram to illustrate it, which shows that the template plasmid is not linearized. The diagram and data have been moved to the Fig. 1f.

9. Figure 2a; It would be helpful if the authors change the relative color of I-SceI, PacI and NcoI sites in the Figures since NcoI site on P allele seems to be a mutation site with white color (but not on Neo+ allele).

As requested by the reviewer, we have changed the colors and have made the I-SceI and PacI mutation sites black.

10. Figure S6a; a gel is too dark to see RAD51 band.

We were unable to repeat this gel (now in Fig. S8d), but the band is visible, especially when the file enlarged.

11. Figure S6f; “Swsap1” in the right could be mouse *Swsap1*. If so, the caption should be changed to clarify.

We think the reviewer is referring to the “Swap” mutant, in which human Walker A motif is now changed to the mouse Walker A motif. We clarified this in the Fig. S6f legend by specifying that there are 4 amino acid changes.

12. Figure 3e and S6g; Although different Y axes are used, the results could come from the same experiment. However, why P-values are different between the two graphs.

We agree this is confusing. Data shown in what is now Fig 2d normalizes values relative to wild-type cells within each experiment to account for variability between experiments, while the data in the previous Fig. S6g is the raw data. To avoid confusion, we have removed this panel (Fig. S6g), although the raw data is still provided in the raw data file.

13. Page 8, second paragraph and Figure 3g; BRC3 reduced IH-HR in *swsap1*^{-/-} cells. If so, please show the P-value in the graph.

BRC3 reduced IH-HR in *Swsap1*^{-/-} cells (0.16 to 0.1, Fig. 2e), however, it is not significant and so the P-value is not shown.

14. Page 9, second paragraph: in this LOH assay in the bottom of Figure 3b, ratios (percent) are shown. It would be nice to show actual frequencies of LOH in the absence and presence of Tet and to check statistical differences.

Thank you for this suggestion. We have now included this data in Fig. 3b, as requested, with a breakdown in Supplementary Fig. 9b. A reduction in LOH frequency is observed upon loss of SWS1 or SWSAP1, either in the presence or absence of BLM, consistent with our model that the complex promotes the formation of recombination intermediates that can undergo crossing over (Fig. 4d).

Reviewer #2 (Remarks to the Author):

Prakash et al. provide interesting and valuable new information regarding the role of the SWS1-SWSAP1-SPIDR complex in homology-directed repair (HDR) and reveal a novel function of this complex in promoting inter-homolog homologous recombination (IH-HR). They show that SWS1, SWSAP1, and SPIDR are required for meiosis and RAD51 and DMC1 focus formation in spermatocytes. In addition, they observe a reduced level of RAD51 focus formation upon IR irradiation, but cells deficient for SWS1-SWSAP1-SPIDR perform gene conversion type of HR in the context of the DR-GFP assay at normal levels. Interestingly, the absence of any component of this complex results in decreased IH-HR as measured using the S/P reporter. These results are in contrast to more general disruptions of HDR by the RAD51-KR mutant or a BRCA2 peptide (i.e., both decreased signal in the DR-GFP assay or survival using the S/P reporter). Interestingly, the role of SWS1-SWSAP1-SPIDR in IH-HR appears to be coupled to BLM function. Depletion or inhibition of BLM results in the previously reported elevated sister chromatid exchange (SCE) phenotype, but co-depletion of SWS1-SWSAP1-SPIDR and BLM leads to WT levels of SCEs. The authors propose a model in which the SWS1-SWSAP1-SPIDR complex stabilizes recombination intermediates that are resolved by BLM. In the absence of BLM activity (such as in Bloom syndrome), SWS1-SWSAP1-SPIDR promote SCEs and crossovers because BLM is no longer available to regulate these activities.

The manuscript is well written and the results are impactful. Some small changes are recommended.

1. Figure 1A and Supp. Fig 1 do not add much and implies SPIDR interacts with both SWS1 and SWSAP1 despite the experimental data only showing SWSAP1-SPIDR (Supp. Fig 1G).

Thank you for pointing this out. We have now added data with purified proteins to Fig. 1a showing that the three proteins, SWS1, SWSAP1 and SPIDR, form a complex.

2. Would the depletion of SWSAP1 result in a mild hyper-recombination phenotype based on the results in Fig. 1F?

This is likely due to experimental variation between clones (now Fig. 1e and Supplementary Fig. 6a, left panel). To look into this further, we performed complementation experiments and saw that expression of SWSAP1 does not reduce HDR in the *Swsap1* mutants (Supplementary Fig. 6a, right panel), supporting clonal variation and not a “mild hyper-recombination phenotype”.

3. Proficient HR despite reduced RAD51 foci after SWS1, SWSAP1, or SPIDR disruption needs to be explained better.

We modified the discussion as follows to more clearly explain this point:

“RAD51 filament formation is clearly required for HDR, since disrupting filaments through BRC3 expression reduces HDR. However, observable RAD51 focus formation, which involves a large protein assembly and so likely has a low threshold for detecting defects, is apparently not required for all types of HDR. In particular, HDR in reporters like DR-GFP typically involves short sequence repeats (<1 kb) and gives rise to short gene conversion tracts (typically <50 bp)⁴³, such that only transient or limited RAD51 nucleoprotein filament formation, below the limit of detection, may be required. Given that *Sws1*, *Swsap1* and *Spidr* mutant mice are viable, while *Brca2* and canonical RAD51 paralog mutants are not²¹, it seems likely that proficient HDR as measured by the DR-GFP reporter reflects physiological intra-chromosomal HDR that is required for survival through embryogenesis, whereas observable RAD51 focus formation is not.”

4. Liu et al (PMID:21965664) showed that formation of the SWS1-SWSAP1 complex is required for stability of the partner. Can the authors inform us about the levels of SWS1, SWSAP1, and SPIDR in the various cell lines? Unfortunately, the commercially available antibodies are not reliable enough for detecting the endogenous cellular proteins in cells, so we have not been able to test this.

5. In conjunction with the IH-HR result in Supp. Fig 6, does mutation of the SWSAP1 Walker A or B box affect RAD51 focus formation or gene conversion in the DR-GFP assay?

As with IH-HR, the SWSAP1 Walker A or B box mutants do not affect gene conversion in the DR-GFP assay. We have now added this data to Supplementary Fig. 7g.

Reviewer #3 (Remarks to the Author):

Summary

In this manuscript, Prakash et al. describe the importance of the SWS1-SWSAP1-SPIDR complex in the control of distinct modes of homologous recombination. This complex is known to play a role in the control of inter-homolog recombination during meiosis. The authors find that this triple S complex also regulates inter-homolog recombination during mitotic HDR, as measured by a genetic assay in human cells. Intra-chromosomal recombination as measured by the DR-GFP assay is largely unaffected. Strikingly, the high rate of SCEs in BLM^{-/-} cells that gives rise to the harlequin chromosome phenotype is rescued upon depletion of components of the SSS complex. The authors propose a model whereby the SSS complex stabilizes RAD51 filaments.

Overall, the experiments presented in the manuscript are well-designed and the data is of high quality. The authors have an abundance of genetic evidence for the role of the SSS complex in regulating inter-homolog recombination in mitosis; additionally, the SCE phenotype in BLM cells is interesting and of potential therapeutic interest. The novelty of this work is high due to the fact that ablation of any of the SSS complex components largely impairs inter-homolog recombination but not intra-homolog recombination, which is a novel finding in the context of mitotic cells. This “separation of function”-like phenotype is not possible by ablation of RAD51, which impairs recombination generally. However, despite the exciting findings, this work does not provide much mechanistic insights into the role of SWS1-SWSAP1-SPRN complex in IH-HR recombination, which at this stage remains somewhat speculative.

A major role for the SWS1-SWSAP1-SPRN complex in regulating RAD51 foci formation during meiosis (IH-HR) has already been established by the same group in a previous work (Abreu et al. 2018, Nature Communications). The lack of a mechanism connecting the SWS1-SWSAP1-SPRN complex and RAD51 dynamics during IH-HR is an important gap in the paper that somewhat reduces its impact and significance. For this reason, the authors should try to better characterize the role and mechanism by which the SWS1-SWSAP1-SPRN complex stabilizes the RAD51 nucleofilament, particularly in the context of BLM deficient cells. The employment of in vitro assays to show the effect of the SSS complex on RAD51 filament stability could be useful (Wang et al. 2015, Molecular Cell: doi: 10.1016/j.molcel.2015.07.009), although that may be beyond the scope of this publication.

We are keenly interested in the biochemical/biophysical activities of this three member complex on RAD51, but this is significantly more intense to investigate than a single point mutation of RAD51 itself, as in Wang et al. It is also noteworthy in this regard that biochemistry on the canonical RAD51 paralog complexes (RAD51BCDX2,CX3) has been extraordinarily difficult despite their initial purification 20 years ago. Given the extensive genetic interrogation of HDR we provide in the manuscript, we are grateful that the reviewer understands that biochemical interrogation of SWS1-SWSAP1-SPIDR is beyond the scope of our manuscript.

Moreover, we believe our genetic experiments provide substantial mechanistic insights which would be difficult to discern from biochemical assays. For example, we are able to distinguish genetic requirements for intra and interchromosomal HDR (IH-HR) for which no biochemical assays have been developed. In addition, we find that observable RAD51 focus formation tracks with IH-HR proficiency. Notably, hyper-stabilizing RAD51 filaments by expressing RAD51 K133R does not restore IH-HR in SSS mutants. RAD51 K133R is defective in ATP hydrolysis, such that filaments are not able to turnover. In our new data, we show that another RAD51 mutant, RAD51 I287T, which stabilizes filaments by another mechanism (increased DNA binding), is also unable to restore IH-HR in SSS mutants, nor is gross overexpression of RAD51 through transient transfection. By contrast, again in new data we provide, doubling the amount of RAD51 WT leads to an increase in IH-HR, albeit partial, which supports the idea that longer RAD51 filaments are more proficient in IH-HR. (New data are in Fig. 2f, Supplementary Fig. 7c-f.) We have reorganized some of the discussion to improve the clarity of these points and have added the following sentence:

“Our finding that doubling the amount of RAD51 WT partially suppresses the IH-HR defect when SWS1-SWSAP1-SPIDR is deficient is consistent with the complex promoting the stable assembly of longer RAD51 nucleoprotein filaments (Fig. 4d).”

Other Major Concerns

1- It was confusing whether the authors consider SCEs as intra-chromosomal HDR events or not. SCEs are certainly not inter-homolog HDR events, but whether they are intra-chromosomal or not may be a matter of

semantics, or if someone considers chromatids on a replicating chromosome as part of the same chromosome. The authors should clarify their view on this, especially since they do observe a reduction in SCEs upon SSS loss of function. Therefore SCEs are not really behaving as intra-chromosomal HDRs as measured by DR-GFP, but seem to behave as inter-homolog HDR events.

The reviewer raises an interesting point. To clarify, we consider both SCEs and DR-GFP to reflect intra-chromosomal HDR: SCEs involve sister chromatids, while HDR in DR-GFP can be inter-sister or intra-chromatid. (Note: intersister events involving direct repeats have previously been verified by us and Ralph Scully's lab, although we cannot differentiate them using the DR-GFP reporter.) The primary difference is that SCEs involve crossing-over, while HDR measured by the DR-GFP reporter does not, in that all GFP+ events are noncrossovers. To clarify this point, we added the following sentence when DR-GFP is first mentioned in the results:

“To directly measure HDR, we utilized the commonly employed reporter DR-GFP, which measures HDR without crossing over between direct repeats (intrachromatid or sister chromatid) leading to GFP positive cells following I-SceI endonuclease expression²³.”

We also modified a sentence in the discussion to clarify:

“By contrast, HDR as assayed between direct repeats (intra-chromosomal HDR), which can reflect HDR within a chromatid or between sister chromatids⁵⁶, is not reduced in *Sws1*^{-/-}, *Swsap1*^{-/-}, or *Spidr*^{-/-} mutant mouse fibroblasts or ES cells.”

2- It would be useful and informative that the authors test if depletion of BLM rescues the Rad51 foci reduction in SSS complex depleted cells. This would better characterize the mutually antagonistic effects of BLM and SSS proteins.

While depletion of BLM does not affect RAD51 focus on its own, when we knocked down BLM in *Swsap1* mutant cells, we found a partial restoration of RAD51 focus formation, as predicted by the reviewer (Fig. 3e) and consistent with a mutual antagonism in this context.

Moreover, in the discussion the authors speculate that X-shaped recombination intermediates are elevated in BLM cells, based on evidence from yeast. According to the yeast model, this phenotype would be “rescuable” by depletion of members of the SSS complex. The authors could test this.

We agree that this would be very interesting to do but we are not aware of similar experiments being performed in mammalian cells and it is beyond the scope of the work presented here.

3- Genetic ablation is used to great effect in this study, but the authors should also test if ectopic overexpression of the SSS complex produces a higher number of SCEs in cells treated with MMS or PARPi. The same should be done for BLM^{-/-} cell lines.

This is a difficult experiment to do because it would involve generating stable cell lines overexpressing 3 different proteins (SWS1, SWSAP1, and SPIDR). This is doable but would take a lot of resources to assess expression of each complex member in numerous clones. Ideally, we would construct such cell lines, but it is a lower priority experiment at the current time and we cannot afford to expend the resources.

4- In Figure 1F and 1G the authors first show no reduction in classical HDR efficiency as measured by DR-GFP in *SWS1*^{-/-}, *SWSAP1*^{-/-} and *SPDR*^{-/-} cells and then, in Figure 1J, they show a great reduction in RAD51 foci formation after IR. This is an intriguing and relevant information to the field. The authors do provide some explanation to this disconnect in the discussion, but it would be useful to go a bit further. Is it just that less stable RAD51 filaments contain less RAD51 molecules and therefore provide lower signal for detection?

The reviewer highlights one of the most surprising parts of our study. As suggested, we have further discussed this point by spelling out our view that shorter/less stable Rad51 filaments, below the level of detection, are sufficient for many HDR events. (end of page 11, top of page 12)

5- The authors should more rigorously test the possibility that disruption of SSS complex components alters resection dynamics. Currently their only evidence to the contrary is the data presented in figure 2C, which considers only one SSS complex component in the presence of a single drug, NCS.

We have now included RPA staining after IR, which shows that there is no significant difference between control and all three mutants for RPA foci (Supplementary Fig. 6h), indicating there is not a resection defect.

6- Authors should include analysis of gene targeting and RAD51 foci also in SWS1^{-/-} and SPDR^{-/-} cells (Figure 1G, K). Same for figure 2D, E and F.

Figure 1G: Gene targeting results were added for Sws1 and Spidr mutant cells, with similar results to the Swsap1 mutant in that there was no discernible effect (former Fig. 1g, now Fig. 1f).

Figure 1K: The time course of RAD51 focus formation shown for Swsap1 mutant cells (former Fig. 1k, now Fig. 1i) provided a gauge for when to examine the other mutants. It was not repeated for the Sws1 and Spidr mutant cells, since they also show a reduction in RAD51 foci at the 2 hr time point (Fig. 1j).

Figure 2D is specific to SWSAP1 (long and short forms of the human SWSAP1 protein) so there is nothing to compare in this figure panel for either SWS1 or SPIDR. Note that human SWS1 was tested in Supplementary Fig. 7b and found to complement the mouse mutant. We have not tested human SPIDR in mouse cells but SPIDR has a similar level of conservation between mouse and human (see Supplementary Fig. 1 legend), so it is likely that human SPIDR will be functional in mouse cells.

Figure 2E,F: The primary purpose of these panels is to show that the RAD51 KR mutant and the BRC repeat suppress HDR measured by both the DR-GFP and IH-HR reporters, unlike the Sws1, Swsap1, and Spidr mutants which are only defective in IH-HR. However, to complement the results we showed with the Swsap1 mutant, we expressed both the RAD51 KR mutant and the BRC repeat in Sws1 mutant cells and obtained no significant reduction in DR-GFP HDR, indicating that SWSAP1 and SWS1 do not provide a “back-up” for DR=GFP HDR when it is compromised by disruption of the principle HDR factors. The new data are added to Supplementary Fig. 7h.

7- I felt that the introduction lacks a better background on SSS proteins. A more in-depth description of their domain organization and some of the previously defined activities of these proteins would help better situate the reader who is unfamiliar with the topic.

Regarding domain structure, we added both the “structure” of the Zn-coordinating domain (CxC(x₁₅)CxH) and RAD51 interaction motif (RxxA) to the introduction. SWSAP1 also has an identified domain with Walker A,B boxes which is diagrammed in Supplementary Fig. 8b in reference to mutations we create. SPIDR does not have any identified domains. For each protein, we refer readers to Supplementary Fig 1 which highlights features of each protein. We also now mention the biochemistry done on the diverged yeast complex – a 4 member complex with 3 RAD51 paralogs rather than 1 like the mammalian complex - to help orient readers.

8- The use of BRC3 peptide and the RAD51 KR mutant is confusing. In the case of RAD51-KR mutant, if it indeed leads to hyper-stabilized RAD51-filaments, why doesn't it improve inter-homolog HR? This should be

the case especially in SWS1-depleted cells. Also, my expectation is that the BRC3 peptide should exacerbate the phenotypes of SWS1-depleted cells. But this is not the case, and the authors should clarify.

RAD51 K133R: The hyper-stabilization of RAD51 filaments containing RAD51 K133R means that the filaments cannot turnover so HDR cannot be completed (in DR-GFP or IH-HR).

BRC3: BRC3 expression significantly reduces DR-GFP recombination in wild-type and mutant cells (Fig. 2d, Supplementary Fig. 7b). The fact that it does not further exacerbate HDR in DR-GFP in the mutants is in keeping with the lack of a role for these proteins in this type of HDR. The filaments that are present in the mutants may be as disrupted as much as those in wild-type cells. However, it appears that BRC3 further exacerbates the IH-HR deficiency, although this reduction is not statistically significant (Fig. 2e). In IH-HR, we are hypothesizing that longer, more stable filaments are required, so BRC3 may have a greater disruptive role in the mutant.

Minor Concerns

1-Figure 1L is confusing and the connection between RAD51 filaments, viability, and fertility is unclear. The authors should simplify the schematic in the form of a flowchart akin to what is described in the figure legend or omit the figure entirely.

We removed Fig 1L and incorporated aspects into Fig. 4d and explained the connection between RAD51 filaments, viability, and fertility in the manuscript text in the discussion section.

2- Figure S1F is misleading. Is this a CoIP? Which is the protein immunoprecipitated? Which bands corresponds to SWS1 and which corresponds to SWSAP1? Where is Flag-SWS1 in the input sample? Same for Figure S1G. Which is the protein immunoprecipitated? For consistency, please show molecular weight of the indicated proteins.

We have now included all the controls and have labelled them appropriately and all the gels show molecular weight of the indicated proteins.

3- Figure S1H: Where are the Flag-SWS1 and FLAG-SWAP1 western blot signals? It looks like SWSAP1 interacts more strongly with RAD51 than SWS1 but is harder to say without an IP control. Authors need to show all the IP controls.

We have now included all the controls and have labelled them appropriately and all the gels show molecular weight of the indicated proteins.

4- In many cases, the information in figures needs to be improved. For example, it is often not clear what is being measured and how. For example, the Y axis title for the bar chart in Figure 2B reads “relative to WT.” It should read something along the lines of “IH-HR efficiency (relative to WT)” to indicate more clearly what assay/measurement is being used. Conversely, many of the figure legends are too verbose and include interpretations of the data that is best left for the results/discussion section.

Thank you for the suggestion. We have clarified what we are measuring both on the Figures (e.g., “relative to WT” is now “Relative to WT cells”) and have provided more detailed explanations in the Figure legends (e.g., “Relative to WT cells, colony counts expressed relative to wild-type cells transfected with an empty vector within each experiment.”)

REVIEWERS' COMMENTS

Reviewer #1 (Remarks to the Author):

Review on Prakash et al.
NCOMMS-20-19537A

In the revised version of the paper by Prakash et al., the authors answered most of comments by reviewers in a proper way. The paper shows a role of a new RAD51 mediator complex, SWS1-SWSAP1-SPIDER, in inter-homolog recombination such as meiotic recombination and sister chromatid exchange and its relationship with BLM helicase in human and mouse with fruitful data sets. These are very wonderful works with thorough characterization of the complex in various homologous recombination (HR) pathways, which provides a new insight on the regulation on HR pathways in mammalian cells. However, there are several points which the authors need to change so that readers can easily digest the content.

Figure legends: Since, in Result sections, the authors mainly mentioned a conclusion of each experiments with minimum wording. The authors should mention in more detail what kind of experiments were done. Current description in Result section and Figure Legends is not sufficient for readers to follow the results with fair evaluation. For example, Fig. 1A shows complex formation of SWS1-SWSAP1 with SPIDR with "IP-western of purified proteins using amylose resin with a physiological salt condition of 100mM NaCl". It would be nice for readers to see brief description on the method in Legends.

Fig.1c,d, legend: Please explain what is red (SYCP3) and how SYCP3 look like in early zygonema.
Fig.2c, legend: please explain how RPA-GFP was introduced into MEF. It would be nice to show an image of representative tracking in Supplementary Fig.
Line 223: What is a nature of RPA-GFP; is this Rfa1-GFP?

The authors may need to pay attention to the consistency on how to show the data. For DR-GFP assay, Fig.1e shows % of GFP+ while Fig.2d shows relative to WT.

Line 161: Please add 6a "left".

Line 176: Supplementary Fig.6"d,e" should be 6"c,d".

Line 176-177: this sentence is almost identical to a previous one. Please rephrase this.

Line 331: Please explain a BLM inhibitor in more detail and add the reference.

Line 336: Supplementary Fig.9c is not for RAD51 focus assay. Replace the graph or image and add Fig.9C-SCE in line 333, to show the inhibitor works as a control.

Line 340: Fig. 3"e" should be 3"f".

Line 344: Fig. 3"f" should be 3"g".

Fig. 1c,d, legend: See also Supplementary Fig."3"e,f should be "4"e.f.

Fig. 2f: Remove "/swsap1 cells" in Y axis-Relative to WT cells.

Fig. 3b: Please add label "b". It would be nicer to draw of a schematic figure with the distal marker based on the figure in Fig. 2a to avoid misunderstanding.

Fig 4d (and in the text): What is difference between HDR and HR? The authors may use either one for consistency.

Supplementary Figure 1g: Please add a schematic complex with RAD51 and SWSAP1, which was shown in this experiment, in addition to RAD51-SWSAP1-SWS1 complex

Supplementary Figure 6b right: In legend, please explain how the authors calculated % HDR in the graph.

Supplementary Figure 7: Please add "relative to WT cells" as Y-axis caption.

Supplementary Figure 9a: An image of sws1-/- spread may show different magnification compared to other ones, given longer chromosomes.

Reviewer #2 (Remarks to the Author):

Authors have done a very fine job addressing my points.

The manuscript documents impactful findings that shed light on how the trimeric SWS1-SWSAP1-SPIDR complex promotes IH type of recombination events. The manuscript is also exceptionally well written for the general readership.

I enthusiastically support the publication of this study in Nature Communications.

Reviewer #3 (Remarks to the Author):

I am satisfied with the responses to my concerns and with the revised version of the manuscript.

REVIEWERS' COMMENTS

Reviewer #1 (Remarks to the Author):

Review on Prakash et al.
NCOMMS-20-19537A

In the revised version of the paper by Prakash et al., the authors answered most of comments by reviewers in a proper way. The paper shows a role of a new RAD51 mediator complex, SWS1-SWSAP1-SPIDER, in inter-homolog recombination such as meiotic recombination and sister chromatid exchange and its relationship with BLM helicase in human and mouse with fruitful data sets. These are very wonderful works with thorough characterization of the complex in various homologous recombination (HR) pathways, which provides a new insight on the regulation on HR pathways in mammalian cells. However, there are several points which the authors need to change so that readers can easily digest the content.

Figure legends: Since, in Result sections, the authors mainly mentioned a conclusion of each experiments with minimum wording. The authors should mention in more detail what kind of experiments were done. Current description in Result section and Figure Legends is not sufficient for readers to follow the results with fair evaluation. For example, Fig. 1A shows complex formation of SWS1-SWSAP1 with SPIDR with “IP-western of purified proteins using amylose resin with a physiological salt condition of 100mM NaCl”. It would be nice for readers to see brief description on the method in Legends.

We thank the reviewer for this suggestion and have so modified the legend as suggested.

Fig.1c,d, legend: Please explain what is red (SYCP3) and how SYCP3 look like in early zygonema.

We have modified the legend to explain that SYCP3 is part of the developing chromosome axes: Foci were counted on the developing chromosome axes that are marked by SYCP3, which at early zygonema form short linear stretches.

Fig.2c, legend: please explain how RPA-GFP was introduced into MEF. It would be nice to show an image of representative tracking in Supplementary Fig.

Line 223: What is a nature of RPA-GFP; is this Rfa1-GFP?

We have modified the legend to specify that it is RPA32-GFP. The remaining details are in the Methods.

The authors may need to pay attention to the consistency on how to show the data. For DR-GFP assay, Fig.1e shows % of GFP+ while Fig.2d shows relative to WT.

In Figure 1e we presented values as % GFP, since this was the first time these experiments were presented and % GFP gives readers a sense of the absolute numbers and variation between experiments. However, subsequently we presented it relative to WT to minimize the variation between experiments. For example, in some experiments cells uniformly transfected less well. (Note: Raw data are available in supplemental data file.)

The following minor points, many of which were on the extended comments page, were also addressed.

Line 161: Please add 6a “left”.

Line 176: Supplementary Fig.6”d,e” should be 6”c,d”.

Line 176-177: this sentence is almost identical to a previous one. Please rephrase this.

Line 331: Please explain a BLM inhibitor in more detail and add the reference.

Line 336: Supplementary Fig.9c is not for RAD51 focus assay. Replace the graph or image and add Fig.9C-SCE in line 333, to show the inhibitor works as a control.

Line 340: Fig. 3”e” should be 3”f”.

Line 344: Fig. 3”f” should be 3”g”.

Fig. 1c,d, legend: See also Supplementary Fig.”3”e,f should be “4”e.f.

Fig. 2f: Remove “/swsap1 cells” in Y axis-Relative to WT cells.

Fig. 3b: Please add label “b”. It would be nicer to draw of a schematic figure with the distal marker based on the figure in Fig. 2a to avoid misunderstanding.

Fig 4d (and in the text): What is difference between HDR and HR? The authors may use either one for consistency.

Supplementary Figure 1g: Please add a schematic complex with RAD51 and SWSAP1, which was shown in this experiment, in addition to RAD51-SWSAP1-SWS1 complex

Supplementary Figure 6b right: In legend, please explain how the authors calculated % HDR in the graph.

Supplementary Figure 7: Please add “relative to WT cells” as Y-axis caption.

Supplementary Figure 9a: An image of sws1-/- spread may show different magnification compared to other ones, given longer chromosomes.

Reviewer #2 (Remarks to the Author):

Authors have done a very fine job addressing my points.

The manuscript documents impactful findings that shed light on how the trimeric SWS1-SWSAP1-SPIDR complex promotes IH type of recombination events. The manuscript is also exceptionally well written for the general readership.

I enthusiastically support the publication of this study in Nature Communications.

Reviewer #3 (Remarks to the Author):

I am satisfied with the responses to my concerns and with the revised version of the manuscript.